# Nonstoichiometric Strontium Ferromolybdate as an Electrode Material for Solid Oxide Fuel Cells

Gunnar Suchaneck [1,*] and Evgenii Artiukh [1,2]

[1] Solid-State Electronics Laboratory, TU Dresden, 01062 Dresden, Germany
[2] SSPA "Scientific-Practical Materials Research Centre of NAS of Belarus", Cryogenic Research Division, 220072 Minsk, Belarus
* Correspondence: gunnar.suchaneck@tu-dresden.de

**Abstract:** This review is devoted to the application of $Sr_2FeMoO_{6-\delta}$ (SFM) and $Sr_2F_{1.5}Mo_{0.5}O_{6-\delta}$ ($SF_{1.5}M$) in $La_{1-x}Sr_xGa_{1-y}Mg_yO_{3-\delta}$ (LSGM)-based SOFCs. We consider the most relevant physical properties (crystal structure, thermodynamic stability, iron and molybdenum valence states, oxygen vacancy formation and oxygen non-stoichiometry, electrical conductivity), A- and B-site ion substitution, and the performance of $SF_{1+x}M$ SOFCs (polarization resistance, operation with hydrogen, operation with hydrocarbons and methanol). Their properties can be tailored to a particular application by the substitution of different metal cations into their lattices. $SF_{1+x}M$ materials are excellent catalysts in hydrocarbon oxidation and can prevent carbon deposition due to the ability to exchange lattice oxygen with the gaseous phase. Moreover, they are sulfur tolerant. This opens the way to direct hydrocarbon-fueled SOFCs, eliminating the need for external fuel reforming and sulfur removal components. Such SOFCs can be greatly simplified and operate with much higher overall efficiency, thus contributing to the solution to the lack of energy problem in our modern world.

**Keywords:** solid oxide fuel cells; strontium ferromolybdate; SOFC performance

## 1. Introduction

Solid oxide fuel cells (SOFCs) convert chemical energy into electrical energy by oxidizing fuel in an electrochemical device. They are considered one of the most important power generation technologies because of their efficiency beyond 60%. SOFCs can utilize a wide variety of fuels, such as $H_2$, natural gas, liquid fuels, biofuels and gasified coal, with a relatively low sensitivity to fuel impurities compared with other types of fuel cells [1–3]. Current research focuses on the development of intermediate-temperature solid oxide fuel cells (IT-SOFCs), which are operated within 500–800 °C [4,5]. Lowering the operating temperature of high-temperature SOFCs suppresses component degradation, extends the range of acceptable materials, serves to improve cell durability and reduces the system cost [6]. However, as the temperature is reduced, the catalytic activity of cathode materials is reduced as well, leading to sluggish electrode kinetics for the oxygen reduction reaction and resulting in large interfacial polarization resistances [7]. This represents a barrier to IT-SOFCs' commercialization.

A SOFC device usually consists of an electrolyte and two porous electrodes, an anode and a cathode. At the cathode, oxygen ($O_2$) is electrochemically reduced to two oxygen ions, $O^{2-}$. At the anode, $O^{2-}$ reacts with the fuel $H_2$ or hydrocarbons yielding $H_2O$ or $CO_2$, respectively, thereby releasing electrons. The electrolyte conducts oxygen ions $O^{2-}$ while its electronic conductivity should be kept as low as possible to prevent leakage currents. Many materials have already been used to make conventional SOFCs, including perovskite-type $ABO_3$ oxides and fluorites [4]. Double perovskites have been the subject of two recent reviews [8,9]. In the case of hydrocarbon fuels, double perovskites have attracted our attention due to the beneficial catalytic activity for methane oxidation reaching 80% at 530 °C [10]. In the following, we use the following notations for chemical elements, which are all

introduced by corresponding chemical formulas: B—barium, Ca—calcium, C—cobalt, Ce—cerium, F—iron, G—gallium, Gd—gadolinium, L—lanthanum, Mn—manganum, Mg—magnesium, M—molybdenum, N—niobium, Ni—nickel, O—oxygen, P—praseodym, S—strontium, Sc—scandium, Sm—samarium, Y—yttrium and Z—zirconium. No fractions of the element are given, e.g., B represents Ba2, Ba2−xBax and Ba1−x, and O represents On-$\delta$ with n = 2,3,6. Note that we are considering double perovskites A2BB'O6-$\delta$, where the total number of ions at both A-site and B-site should be two.

Recently, a symmetrical SOFC has been reported using $La_{0.75}Sr_{0.25}Cr_{0.5}Mn_{0.5}O_{3-\delta}$ (LSCrMnO) as a redox (reduction-oxidation) stable material for both the anode and cathode. A redox-stable cathode is also advantageous for traditional SOFCs since some leakage might occur, and fuel is introduced into the cathode side [11]. In symmetrical SOFCs, sulfur poisoning or coke formation on the surface of the anode is eliminated, reversing fuel gas and air flow. The oxidant (typically air) flushed any sulfur or carbon species absorbed on the electrode, thereby regenerating the electrode from sulfur or coke deactivation. Another advantage of symmetrical SOFCs with redox-stable electrodes is the enhancement of the cathode durability since the $O_2$ partial pressure at the cathode triple phase boundary region can be quite low when SOFCs are operated at a low voltage. Moreover, a redox-stable electrode is a better choice in the case of some leakage [12]. In symmetrical SOFCs, the number of fuel cell components reduces from at least three materials to just two simplifying the production process, reducing manufacturing cost as well as improving thermal compatibility because only one type of interface is present. Nevertheless, materials requirements are much higher since electrodes for symmetrical cells must simultaneously maintain a stable structure and sufficient electrical conductivity in both air and fuel gas atmospheres. Moreover, symmetrical SOFCs provide balanced stresses and, thus, favorable mechanical properties for vibration and thermal cycling.

Typical SOFC cathode materials are transition metal-based perovskites such as $La_{0.8}Sr_{0.2}MnO_3$ (LSMn), $La_{0.8}Sr_{0.2}Fe_{0.8}Co_{0.2}O_{3-\delta}$ (LSCF) and $Ba_{0.5}Sr_{0.5}Co_{0.8}Fe_{0.2}O_{3-\delta}$ (BSCF) [6]. Commonly, LSMn is applied as a composite with yttria-stabilized zirconia (YSZ) electrolyte. This extends the triple-phase boundary, which is an active site for oxygen reduction and increases ionic conductivity [5,6]. LSMn has a high electrochemical activity for the $O_2$ reduction reaction at high temperatures, good thermal stability and chemical stability, and matches well with the electrolyte. Nevertheless, LSMn is a poor ionic conductor making it not suitable for intermediate-temperature SOFC operation [8]. The reason for this disadvantage is the absence of a sufficient number of oxygen vacancies [6]. $Sr^{2+}$ doping is carried out at the $La^{3+}$ site to introduce oxygen vacancies in the parent structure of $LaMnO_3$ due to the charge compensation mechanism [5]. On the other hand, the strontium doping on lanthanum introduces extra holes in the valence band of the p-type material and thus increases electronic conductivity. BSCF is an excellent oxygen permeation membrane material. Despite its excellent electrochemical performance, it has a high thermal expansion coefficient (TEC) of $20 \times 10^{-6}$ $K^{-1}$ between 50 and 1000 °C [6]. LSCF does not react with ceria-based electrolytes and shows good electrical conductivity, high oxygen surface exchange coefficient and good oxygen self-diffusion coefficient between 600 and 800 °C. Its TEC may be lowered by A-site deficiency up to $13.8 \times 10^{-6}$ $K^{-1}$ for $La_{0.6}Sr_{0.2}Co_{0.2}Fe_{0.8}O_{3-\delta}$ at 700 °C, matching with commonly used electrolytes [6]. Recently, a new cathode material, $Ba_{0.9}Co_{0.7}Fe_{0.2}Nb_{0.1}O_{3-\delta}$ (BCFN), was proposed for application with $La_{0.8}Sr_{0.2}Ga_{0.83}Mg_{0.17}O_{3-\delta}$ (LSGM) electrolyte in order to increase cathode stability [13]. For application as cobalt-free, redox stable and electrically conductive IT-SOFC interconnector, lanthanum strontium ferrite was doped with Sc yielding $La_{0.6}Sr_{0.4}Fe_{0.9}Sc_{0.1}O_{3-\delta}$ (LSFSc$_{0.1}$) [14]. Moreover, $SmBaCo_2O_{5+x}$ (SmBC$_2$) was considered as an IT-SOFC cathode [5]. LSCF and SmBC$_2$ are mixed ionic electronic conductors (MIEC), permitting both oxide ion and electron mobility within the cathode material [4,5,8]. MIECs promote oxygen reduction by enhancing the charge transfer process and expanding the active reaction sites beyond the cathode/electrolyte physical interfaces.

SOFC electrolytes are oxygen ion-conducting ceramics. Yttria-stabilized $Y_{1-x}Zr_xO_{2-\delta}$, $x = 8$–10 mol %, YSZ) is the most used SOFC solid electrolyte due to its high chemical stability. It is chemically stable in a wide range of temperatures and oxygen partial pressures and is thermally matched to other SOFC components. However, its ionic conductivity below 800 °C becomes less than $3 \times 10^{-2}$ S cm$^{-1}$ [15]. When the SOFC operating temperature is lowered, an electrolyte material is required which possesses a high enough ionic conductivity. Strontium- and magnesium-doped lanthanum gallate $La_{1-x}Sr_xGa_{1-y}Mg_yO_3$ with $x = 0.1$–0.2, $y = 0.13$–0.2 (LSGM) exhibits an ionic conductivity of about $8.76 \times 10^{-2}$ S cm$^{-1}$ at 700 °C, negligible electronic conductivity and good chemical stability over a wide range of oxygen partial pressures [16]. Other promising electrolytes for IT-SOFCs are $Ce_{1-x}Gd_xO_{2-\delta}$, $0.1 < x$ 0.2, (Gadolinium doped ceria—GDC); $Ce_{1-x}La_xO_{2-\delta}$, x = 0.4, (Lanthanum doped ceria—LDC); and $Ce_{1-x}Sm_xO_{2-\delta}$, x = 0.2, (Samarium doped ceria—SDC) with conductivity values in the order of a few $10^{-2}$ S cm$^{-1}$ at 700 °C, in some cases slightly higher than LSGM [15]. However, they are mixed ion–electron conductors suffering from a lower open-circuit voltage caused by an electron leakage current. At elevated temperatures above 500 °C, $Ce^{4+}$ ions can be reduced at the anode to $Ce^{3+}$ enhancing electronic conductivity and causing additional lattice expansion.

In this work, the considered $Sr_2Fe_{1.5}Mo_{0.5}O_{6-\delta}$ does not react with SDC even at 1200 °C while the interaction with YSZ electrolyte begins at 1000 °C. In the case of interaction with LSGM, no appreciable shift in the position of the diffraction peaks was observed at different sintering temperatures, and no new diffraction peaks assigned to reaction products were detected, indicating good chemical compatibility between LSGM and $Sr_2Fe_{1.5}Mo_{0.5}O_{6-\delta}$, in the studied temperature range between 800 and 1200 °C [17].

The most common anode material is a nickel/yttria-stabilized (Ni-YSZ) cermet, usually in combination with YSZ electrolytes. The nickel provides electronic percolation at typically ~30 vol %. Ni was chosen, instead of Co and noble metals suitable at high temperatures for economic reasons. Moreover, a thick anode NiO-YSZ substrate and a thin YSZ electrolyte can be co-sintered, followed by in situ reductions in nickel oxide. This way, a good interlocking between the anode and electrolyte is achieved. YSZ provides mechanical support for the Ni particles, prevents coarsening of metallic particles, allows ionic transport and increases chemical as well as thermal compatibility with YSZ electrolyte. Ni-YSZ anodes exhibit excellent electrocatalytic properties for operation in $H_2$ fuel [15]. However, they are susceptible to nickel agglomeration, carbon formation and sulfur poisoning, leading to component degradation. Another undesirable feature of Ni-based compounds is redox instability [18]. In order to prevent reactions between a Ni-containing anode and an LSGM electrolyte, a doped ceria-based interlayer, e.g., SDC, should be placed between the Ni-SDC composite anode and the electrolyte [19].

This review is devoted to the application of $Sr_2FeMoO_{6-\delta}$ (SFM) and $Sr_2Fe_{1+x}Mo_{0.5}O_{6-\delta}$ (SF$_{1+x}$M) in LSGM-based SOFCs. These materials are stable in both oxidizing and reducing atmospheres enabling the use of both not only as an anode but also as a cathode and also in symmetrical SOFCs. With regard to its cubic structure under SOFC operation conditions, a correct notation would be $SrFe_{1-x}Mo_xO_{3-\delta}$. However, in order to avoid reader's confusion, the double perovskite notation $Sr_2Fe_{1+x}Mo_{1-x}O_{3-\delta}$ (SF$_{1+x}$M) is used throughout this work. We consider the most relevant physical properties (crystal structure, thermodynamic stability, iron and molybdenum valence states, oxygen vacancy formation and oxygen nonstoichiometry, electrical conductivity), A- and B-site ion substitution and the performance of SF$_{1+x}$M SOFCs (polarization resistance, operation with hydrogen, operation with hydrocarbons and methanol).

## 2. $Sr_2Fe_{1+x}Mo_{1-x}O_{6-\delta}$ Properties

### 2.1. Crystal Structure

The Goldschmidt tolerance factor $t$ [20] is an indicator of the stability and distortion of the crystal structure. In the case of a double perovskite, such as $Sr_2Fe_{1+x}Mo_{1-x}O_{6-\delta}$, an average value of the two B-site radii $(2 - x) \times r_B$ and $x \times r_B{}'$ should be used, yielding

$$t = \frac{r_A + r_O}{\sqrt{2}(\langle r_{BB'} \rangle + r_O)},\tag{1}$$

where $r_A$ and $r_O$ are the ionic radii of the A-site ion and oxygen, and $r_{BB}{}'$ is the average ionic radius of the B-site of an $ABO_3$ perovskite structure. Ionic radii are tabulated for various values of the coordination number, oxidation and spin states [21]. With regard to the $Fe^{3+}$-$Mo^{5+}$ fractions considered in Section 2.4 and assuming a high-spin state of Fe, the tolerance factor $t$ amounts to 1.025, 0.971 and 0.940 for the double perovskites $Ba_2FeMoO_{6-\delta}$ (BFM), SFM and $Ca_2FeMoO_{6-\delta}$ (CaFM), respectively. The lattice symmetries obtained at room temperature are correspondingly cubic, tetragonal and orthorhombic. Assuming the same mixed valence states as in SFM, the value of $t$ for $SF_{1.5}M$ amounts to 0.955. Taking the $Fe^{2+}/Fe^{3+}$ and $Mo^{6+}/Mo^{5+}$ ratios given in [22], the value of $t$ increases slightly to 0.962.

By doping $SrFeO_{3-\delta}$ (SFO) with a certain amount of Mo (about 5%, i.e., $x = 0.95$), doped $SF_{1+x}M$ perovskites can be stabilized in a cubic structure [23–25]. The stabilization of the cubic form occurs due to the incorporation of $Mo^{6+}$, a transition metal with a higher oxidation state than $Fe^{3+}$ and $Fe^{4+}$, leading to a filling of oxygen vacancies by oxygen in order to compensate for the high positive charge of Mo ions [23,26]. A cubic structure at SOFC operation temperatures exhibits better oxygen permeability than other structures, such as hexagonal and rhombohedral [27]. Therefore, it is beneficial to preserve the cubic structure of the electrode material after modification by ion substitution or doping.

The crystal structure of $SF_{1.5}M$ at room temperature refined by Rietveld analysis from X-ray and neutron diffraction was attributed to several space groups: Cubic *Fm-3m* [28–30]; cubic *Pm-3m* with iron and molybdenum disordered on the B-site [25,31]; tetragonal *I4/mcm* of the tetragonal system [32]; and pseudocubic, orthorhombic *Pnma* [33]. These structural differences are due to the tilting of $BO_6$ octahedra. Consequently, different space groups are probably due to different synthesis methods of the samples. Variable temperature, powder X-ray diffraction measurements in the temperature range of 25–800 °C revealed a cubic *Fm-3m* space group without any phase transition [31]. Later, this result was revised, claiming a reversible tetragonal to cubic phase transition occurring between room temperature and 400 °C, both on heating and cooling in either oxygen or hydrogen [32]. A cubic structure was also obtained in $SF_{1.33}M$ [29,34], $Sr_2Fe_{1.4}Ni_{0.1}Mo_{0.5}O_{6-\delta}$ ($SF_{1.4}Ni_{0.1}M$) and $Sr_{1.9}Fe_{1.4}Ni_{0.1}Mo_{0.5}O_{6-\delta}$ ($S_{1.9}F_{1.4}Ni_{0.1}M$) [35]. The cubic structure is associated with a structural B-site disorder. Below $x = 0.2$, the B-site ordering of $SF_{1+x}M$ becomes noticeable, and the crystal structure changes from cubic *Fm-3m* to tetragonal *I4/mmm* [28,36].

In $SF_{1.5}M$, a cubic structure is preserved by A-site substitution of $Sr^{2+}$ by $Ba^{2+}$ up to 30% [37] and $Sr^{2+}$ by $Ca^{2+}$ up to 30% [38], and B-site substitution of $Fe^{2+/3+}$ by $Co^{2+/3+}$ up to 33% [39], $Fe^{2+}/Fe^{3+}$ by $Ni^{2+}$ up to 26.6% [40,41], $Fe^{2+}/Fe^{3+}$ by $Nb^{4+}/Nb^{5+}$ by 6.7% [30], $Fe^{2+}/Fe^{3+}$ by $Sc^{3+}$ up to 13.3% [42] and $Mo^{5+/6+}$ by $Sn^{2+/4+}$ up to 100% [43]. Contrarily, at room temperature, $Sr_{2-x}Ca_xFe_{1.5}MoO_{6-\delta}$ ($S_{2-x}C_xFM$), $x = 0$–0.6, [44] and $Sr_2Fe_{1.5}Mo_{0.5-x}Nb_xO_{6-\delta}$ ($S_2F_{1.5}M_{0.5-x}N_x$), $x = 0$–0.2 [45] were assigned to the space group *Pnma*.

### 2.2. Thermodynamic Stability

The $SrFeO_{3-\delta}$ (SFO) system exists over the oxygen composition range $0.16 \leq \delta \leq 0$ as four distinct compounds with the nominal composition $Sr_nFe_nO_{3n-1}$ ($n = 2, 4, 8$ and $\infty$). The end member $SrFeO_3$ ($n = \infty$) possesses a simple cubic perovskite crystal structure, whereas the oxygen-deficient ($n = 2, 4$ and 8) members each adopt a different vacancy-ordered perovskite crystal structure [46]. In the brownmillerite phase $Sr_2Fe_2O_5$ possessing the maximal oxygen nonstoichiometry ($\delta = 0.5$), oxygen long-range ordering results in a

regular alternation of layers of corner-connected $FeO_4$ tetrahedra and $FeO_6$ octahedra. The high oxygen non-stoichiometry of $Sr_nFe_nO_{3n-1}$ does not translate into a high oxygen ion conductivity. The formation of ordered oxygen vacancies drastically reduces oxide ion conduction, and oxygen deficiency decreases both mobility and carrier concentration [47]. Thus, only SFO (with $n = \infty$) is useful for SOFC applications. Under ordinary pressure, SFO is always oxygen-deficient, leading to a composition of samples slowly cooled in the air in the range of $SrFeO_{2.80}$ to $SrFeO_{2.85}$ [48].

$SFe_{1+x}M$ is a solid solution of SFO with molybdenum substituting for iron. Double perovskite SFM is thermodynamically unstable in air due to insufficient Mo and Fe solubilities in $SrFeO_{3-\delta}$ and $SrMoO_4$ (SMO), respectively [49]. The solubility of SMO in SFO judged from oxygen non-stoichiometry is about 15 to 17 mol % ($Sr_2Fe_{1.40}Mo_{0.60}O_{6-\delta}$ to $Sr_2Fe_{1.32}Mo_{0.68}O_{6-\delta}$) [49,50], that of SFO in SMO amounts to 2.2 mol % [49]. In 1% $H_2/Ar$, this solubility region is increased up to 27 mol % (i.e., up to $Sr_2Fe_{0.92}Mo_{1.08}O_{6-\delta}$, including the stoichiometric composition) [51].

$SFe_{1+x}M$, $0.0 \leq x \leq 0.35$, decomposes into SMO and SFO already at 850 °C, while a reduction at 800 °C in $H_2$—3 % $H_2O$ restores the perovskite phase, whereas $SF_{1.5}M$ was stable after calcining at 1100 °C for 5 h in air [52]. The synthesis of $SFe_{1.5}M$ in the oxidizing environment at 1000 °C was also demonstrated in [12]. On the other hand, SFM could be synthesized only in a reducing environment. According to another report, $S_2F_{1.5}M$ may be synthesized in air, while $SF_{1.33}M$ needs a reducing treatment to form the perovskite structure [29]. $SF_{1.5}M$ was found to be stable in $H_2$ up to 1400 °C [12,17] and in $CO_2$ in the studied range of 600–800 °C [17]. These data indicate that the Mo fraction had a significant influence on the stability of $SFe_{1+x}M$ in air.

The solubility of iron-rich composition $SFe_{1.5}M$ is slightly below maximum SMO solubility in the region of its thermodynamic stability both in air and in a reducing atmosphere of 1% $H_2/Ar$ [12]. On the other hand, $SFe_{1+x}M$ possesses a cubic structure for $0.2 < x < 1$ [31,36,48,53], which in perovskites enhances oxygen permeability [27]. Note that in the compositions range of cubic structures, B-site Fe and Mo ions are highly disordered [31,36], diminishing carrier mobility. This can be overcome by selecting the Mo concentration slightly above the molybdenum percolation threshold in a material exhibiting nanoscale local ordering in a disordered matrix [53]. Experimentally, a percolation threshold value of $x_p = 0.65$ was obtained, while the theoretical value for a homogeneous composite containing randomly oriented spherical fillers of similar size amounts to $x_p = 0.16$ [54]. For ellipsoidal particles, the theoretical $x_p$ value will be even lower since ellipsoidal fillers connect with each other more easily.

Substitution of $Sr^{2+}$ by $Ca^{2+}$ makes $Ca_2FeMoO_6$ (CaFM) thermodynamically more stable than SFM. The $p_{O2}$-region where CaFM is stable increases from $3.2042 \times 10^{-9} \leq p_{O2} \leq 1.6059 \times 10^{-5}$ Pa for SFM to $3.2042 \times 10^{-8} \leq p_{O2} \leq 1.2756 \times 10^{-5}$ Pa for $Ca_2FeMoO_{6-\delta}$ [55]. However, CaFM decomposes in nitrogen above 430 °C into $CaMnO_4$ and $CaFeO_{2.5}$. Therefore, it is incompatible as an anode material in SOFCs.

SFO was observed to be insensitive to water. Alkaline earth metals such as strontium are known to be highly reactive in water-containing atmospheres. As a result, $SF_{1+x}M$ is highly sensitive to water and would decompose into SMO, $Sr(OH)_2$ and FeO with $Sr(OH)_2$, further transforming into $SrCO_3$ by interaction with $CO_2$ [50]. At 800 °C, $SF_{1.5}M$ is stable in water-containing atmospheres. $SF_{1.5}M$ reacts with water at lower temperatures forming $SrMo_4O_{13} \cdot H_2O$, $Fe_3O_4$, SMO, SrO and hydroxides such as $Sr(OH)_2$. However, the $SF_{1.5}M$ phase was restored upon repeated heating to 800 °C [56]. SOFCs containing $SFe_{1.5}M$ possess chemical stability and reasonable performance at the operating temperature [31]. When the SOFC is cooled down, $Sr(OH)_2$ forms, decreasing the conductivity and catalytic activity. When heated back to the operating temperature, reforming of $SFe_{1.5}M$ does not occur due to the sluggish kinetics of solid-state reactions. Moreover, the decomposition products lead to a local increase in volume. This corresponding increase in volume cannot be accommodated by any other means than cracking, generating cumulative damage during cycling [56].

Partial substitution of $Fe^{2+}/Fe^{3+}$ by $Nb^{5+}$ does not affect the formation of the perovskite phase in air. On the other hand, it hinders the reduction of iron to a metallic phase. Thus, $SF_{1.4}N_{0.1}M$ has a more stable structure under a hydrogen atmosphere [30].

### 2.3. Iron and Molybdenum Valence States

SFO has an anomalous valence state, $Fe^{4+}$, which does not appear in the simple Fe-O system [48]. With increasing oxygen nonstoichiometry, the formal charge state on Fe changes from $Fe^{4+}$ in $SrFeO_3$ via a mixture of $Fe^{3+}$ and $Fe^{4+}$ to $Fe^{3+}$ in $SrFeO_{2.5}$ [46]. $Fe^{4+}$ reduction is also favored by heating and/or oxygen pressure decrease. At rather small oxygen pressure values, one can expect a partial reduction in both iron and molybdenum with the formation of $Fe^{2+}$ and $Mo^{5+}$ cations [23]. Another reason for the disappearance of $Fe^{4+}$ and the simultaneous increase in the concentration of the larger $Fe^{3+}$ cations is the incorporation of molybdenum in the crystal lattice. The origin of these effects is the charge compensation of $Mo^{6+}$ ions via the concentration decrease in $Fe^{4+}$ with a simultaneous filling of oxygen vacancy sites by oxygen [23,24].

When the Mo content is low, i.e., $x$ of $SF_{1+x}M$ takes a value of about 0.75, Fe is in the $Fe^{3+}$ oxidation state and Mo adopts a $Mo^{6+}$ state. With increasing Mo content, both the Fe and Mo cations are both partially reduced, resulting in a mixture of $Fe^{3+}$, $Fe^{2+}$, and $Mo^{5+}$ and $Mo^{6+}$ [57]. Here, reduction occurs to keep electro-neutrality. As the $Fe^{2+}/Fe^{3+}$ ratio increases in more Mo-rich compositions, the $Mo^{5+}/Mo^{6+}$ radio shows a peak in $SFe_{1.5}M$. A larger fraction of $Fe^{2+}$ possesses a higher catalytic activity enhancing SOFC performance while the electrical conductivity of SFM decreases [22]. Thus, a balance between the activity, conductivity and stability exists in $SFe_{1.5}M$. Moreover, $SFe_{1.5}M$ was reported to form the percolation path of the Fe-O-Fe bond for the charge carrier to move through the crystal in both hydrogen and air atmospheres [12].

The equilibrium reaction $Fe^{3+} + Mo^{5+} \Leftrightarrow Fe^{2+} + Mo^{6+}$ is directly related to the resistive properties of $SF_{1+x}M$ [51]. The contributions of the $Fe^{2+}-Mo^{6+}$ and $Fe^{3+}-Mo^{5+}$ configurations are quite different among $A_2FeMoO_{6-\delta}$ compounds. The mixed-valence state involves 21–34% $Fe^{3+}$ ($3d^5$; $S = 5/2$)-$Mo^{5+}$ ($4d^1$; $S = 1/2$) and 79–66% $Fe^{2+}$ ($3d^6$; $S = 2$)-$Mo^{6+}$ ($4d^0$; $S = 0$) for SFM [58–61], about 14% $Fe^{3+}$-$Mo^{5+}$ and 86% $Fe^{2+}$-$Mo^{6+}$ for BFM [60] and about 40% $Fe^3$-$Mo^{5+}$ and 60% $Fe^{2+}$-$Mo^{6+}$ for CaFM [60]. Thus, as the A-site cation decreases, the valence balance of $Fe^{3+}+ Mo^{5+} \Leftrightarrow Fe^{2+} + Mo^{6+}$ gradually shifts closer to $Fe^{3+}$-$Mo^{5+}$, and the electronic conductivity increases [60]. It was observed that the contribution of the $Fe^{3+}$-$Mo^{5+}$ configuration to the XPS spectrum of double-perovskites $A_2FeMoO_{6-\delta}$ increases with decreasing A-site cation size [60].

An increase in the A-site cation size by substitution of the $Sr^{2+}$ ion by the larger $Ba^{2+}$ ion gradually shifts the valence balance of $Fe^{2+/3+}$-$Mo^{5+/6+}$ to $Fe^{2+}$-$Mo^{6+}$, which would promote B-site ordering [62,63]. On the other hand, barely any changes in $Fe^{2+}/Fe^{3+}$ and $Mo^{6+}/Mo^{5+}$ couples were obtained in $Sr_{2-x}Ba_xFe_{1.5}Mo_{0.5}O_{6-\delta}$, $x = 0, 0.2, 0.4, 0.6$ ($SB_xFM$) samples [37]. In the latter case, the changes were probably too small to be determined with certainty. Contrarily, $Sr^{2+}$ substitution by $Ca^{2+}$ increases the $Fe^{3+}/Fe^{2+}$ and $Mo^{6+}/Mo^{5+}$ ratios as a whole associated with a reduction in the $\delta$ value, i.e., with a decrease in the oxygen vacancy concentration [44]. A change in the $Fe^{2+}$-$Fe^{3+}$ and $Mo^{6+}$-$Mo^{5+}$ proportion by Ca-doping was also obtained in $SCa_xF_{1.5}M$ [44]. Here, the decrease in $Fe^{2+}$ with Ca doping correlates with the one $Mo^{5+}$ and the increase in $Fe^{3+}$ with $Mo^{6+}$ since oxygen non-stoichiometry also increases. The valences of Fe and Mo change significantly with temperature. The $Fe^{2+}$ and $Mo^{5+}$ content increases with the increase in temperature, influencing the electronic conductivity [31].

At the B-site, $Fe^{2+}$ substitution by $Ni^{2+}$ affects the equilibrium between $Fe^{3+}/Mo^{5+}$ and $Fe^{2+}/Mo^{6+}$. The ratio of $Fe^{2+}/Fe^{3+}$ decreases from 1.46 for $x = 0$ to 0.81 for $x = 0.4$. The $Fe^{2+}/Fe^{3+}$ and $Mo^{6+}/Mo^{5+}$ decrease with Ni fraction, which indicates the amount of $Fe^{3+}$ and $Mo^{5+}$ cations increases. The ratios of the $Fe^{2+}/Mo^{6+}$ and $Fe^{3+}/Mo^{5+}$ redox couples increase with Ni fraction content as x increases. The optimal ratios are attributed to $SF_{1.4}N_{0.1}M$, which is directly related to the conductivity [40]. In another report, the

$Fe^{2+}/Fe^{3+}$ ratio of $SF_{1.5-x}Ni_xM$ decreases from 1.22 at $x = 0$ to 0.97 at $x = 0.1$ and then increases up to 1.56 at $x = 0.3$. Similarly, the $Mo^{6+}/Mo^{5+}$ ratio changes from 1.11 at $x = 0$ to 0.94 at $x = 0.1$ and then up to 1.35 at $x = 0.3$. The difference between the total $Fe^{2+}+Mo^{6+}$ and $Fe^{3+}+Mo^{5+}$ becomes the least when $x = 0.1$ [41]. The contribution of the $Fe^{3+}+Mo^{5+}$ pair to conductivity was further confirmed by its enhancement in Ni-doped $SF_{1.4}Ni_{0.1}M$ cathodes [40] and anodes [41]. For $Sr_2Fe_{1.5}Mo_{0.5-x}Nb_xO_{6-\delta}$ ($SF_{1.5}M_{0.5-x}N_x$), the $Fe^{2+}/Fe^{3+}$ ratio is nearly constant while the $Mo^{6+}/Mo^{5+}$ ratio shows a minimum in the range $x = 0.10$–0.15 coupled with a maximum of the $Nb^{5+}/Nb^{4+}$ ratio at $x = 0.10$ [45]. The substitution of the $Fe^{2+}/Fe^{3+}$-site by high-valence $Nb^{5+}$ induces an obvious reduction from $Fe^{3+}$ to $Fe^{2+}$ to achieve electrical neutrality. The increase in $Fe^{2+}$ (which has a larger radius) results in an expansion of the crystal cell [30].

*2.4. Oxygen Vacancy Formation Energy and Oxygen Non-Stoichiometry*

The formation of an oxygen vacancy $E_f(V_O)$ can be viewed as a process of breaking metal-oxygen bonds followed by the removal of a neutral oxygen atom and the subsequent redistribution of two extra oxygen electrons into the SFM lattice. Cubic $SrFeO_{3-\delta}$, $E_f(V_O)$ exhibits a constant low value (ca. 0.4 eV) for $\delta < 0.05$, increases to ca. 0.5 eV for $0.05 < \delta < 0.1$, and further increases quickly with concentration when $\delta > \approx 0.1$ [64]. This increase is attributed to the local charge redistribution after $V_O$ formation and to overlapping local lattice transformations. In stoichiometric SFM, first principle DFT (density-functional theory, which is a computational quantum mechanical modeling method used in physics) calculations predict that $E_f(V_O)$ along metal–oxygen–metal bonds follows the trend Fe-O-Fe < Fe-O-Mo < Mo-O-Mo. Removing an oxygen atom from Fe-O-Fe costs ∼3.0–3.5 eV, and that from Mo-O-Mo bonds costs ∼4.5–5.0 eV. This is attributed to a much stronger metal–oxide bond of Mo compared to Fe [65]. The Fe−O bonds are easier to break in order to remove oxygen from the lattice to form a vacancy. The formation of an oxygen vacancy consists of not only the removal of a neutral oxygen atom but also the subsequent redistribution of the extra electrons from the oxide ion into the $SF_{1+x}M$ lattice. Therefore, another origin of a low vacancy formation energy is a fully delocalized rearrangement of the extra charge delivered to the lattice upon removal of the neutral oxygen atom. In $SF_{1+x}M$, a partially occupied band with empty states is formed just above the Fermi level by a strong hybridization of the Fe and O states at the conduction band edge. The excess electrons left by the formation of the vacancy easily occupy just these empty states across the Fermi level. Additionally, this effect provides a high conductivity [33]. In fact, DFT calculations for $SFe_{1.5}M$ confirm that $E_f(V_O)$ along Mo-O-Mo bonds is higher than that along Fe-O-Fe bonds at all vacancy concentrations considered. Moreover, $E_f(V_O)$ depends on oxygen non-stoichiometry, and it vanishes at $\delta \approx 0.1$, the value obtained in as-synthesized $SFe_{1.5}M$ material [33]. For $E_f(V_O) \to 0$, the oxygen migration barrier height governs oxide ion diffusion in this material. In summary, excess Fe in $SF_{1+x}M$ provides SOFC materials with higher concentrations of oxygen vacancies that facilitate oxide ion diffusion. Oxygen vacancies in $SF_{1.5}M$ are mainly transported along the Fe-O-Fe bonds and not along the Mo-O-Fe and Mo-O-Mo bonds. At the same time, the Fe-O bonds are relatively weak contributors to a high level of oxygen conductivity. The stronger Mo-O bond leads to a decrease in the value $\delta$ with the incorporation of Mo [25].

The vacancy formation energy of Co-substituted $SF_{1.5-x}Co_xM$ was calculated by means of DFT in [39]. The corresponding values are listed in Table 1. Moreover, $E_f(V_O)$ along Fe-$V_O$–Mo is higher than that along Fe-$V_O$-Fe. It is also higher than $E_f(V_O)$ along Co-$V_O$-Co and along Fe-$V_O$-Co. $E_f(V_O)$ along Fe-$V_O$-Fe exceeds that along Fe-O-Co and all the values of $E_f(V_O)$ decrease with increasing Co content. The latter means that vacancies are much easier to form at higher Co content.

**Table 1.** Oxygen formation energies along different bonds in $SF_{1.5-x}Co_xM$ [39].

| Bond Configuration | $E_f(V_O)$, eV | | | |
|---|---|---|---|---|
| | $x = 0$ | $x = 0.5$ | $x = 1.0$ | $x = 1.5$ |
| Fe-O-Mo | 2.95 | 2.89 | 2.58 | |
| Fe-O-Fe | 2.30 | 2.19 | | |
| Co-O-Mo | | 2.78 | 2.41 | 2.2 |
| Co-O-Co | | | 1.80 | 1.37 |
| Fe-O-Co | | 2.08 | 1.74 | |

DFT usually assumes oxygen-rich conditions where an upper limit of $\mu_O$ is chosen as half of the total energy of a free, isolated $O_2$ molecule in the triplet state at $T = 0$ K [66]. However, this upper limit can never be realized experimentally. On the other hand, the exact oxygen partial pressures in the fabrication processes of thin-film and bulk samples cannot be determined accurately, but their difference in film and bulk sample fabrication should be rather small. Introducing a combined DFT and thermodynamic model and using a temperature- and partial pressure-dependent value of the oxygen chemical potential, $E_f(V_O)$ is reduced from about 4.5 eV in the oxygen-rich limit to about 2.9 eV at $p_{O2} = 2.1 \times 10^4$ Pa (corresponding to air) and to about 1.7 eV at $p_{O2} = 10^{-4}$ Pa (corresponding to an atmosphere of ultrapure, i.e., 99.999 % Ar at 10 Pa) [67].

The lower $E(V_O)$ value in Fe-rich compositions of $SF_{1.5}M$ shifts the homogeneity region to higher values of δ. This is illustrated in Figure 1, which depicts a comparison of the homogeneity regions of both oxygen-deficient $SF_{1.5}M$ and SFM. With the increase in δ, the oxygen permeation and lattice expansion in the reduction atmosphere become much more pronounced [68]. The latter causes larger internal stress to have harmful influences on SOFC operation, such as energy loss, cracking and breakage of the cell.

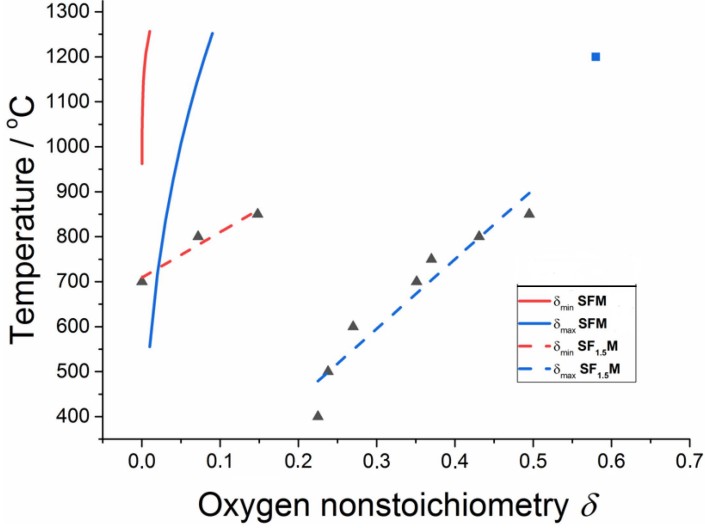

**Figure 1.** Homogeneity region of $SF_{1.5}M$ based on data in refs. Ref. [51] (Blue square) and Ref. [32] (Black triangle) represented in dependence on the lower and upper limits of the oxygen non-stoichiometry parameter δ. The lines are guides for the eye. The homogeneity region of SFM [69] is shown for comparison (solid lines). Adapted with permission from Ref. [69], 2019, Elsevier.

When summarizing crystal structure, thermodynamic stability, Fe and Mo valence state, and oxygen vacancy formation energy, $SF_{1.5}M$ is a compromise that provides a cubic crystal structure, which generates an electronic structure as well as oxygen vacancies that promote suitable electrical transport properties and catalytic activity in oxidizing and reducing atmospheres. With regard to thermodynamic stability considered above, the ideal

Fe to Mo ratio should be around two. The electrocatalytic properties of $SF_{1.5}M$ may be further improved by partial substitution of A-site and B-site ions.

### 2.5. Electrical Conductivity

SFO exhibits the highest conductivity in a temperature range of 400 to 900 °C. Stoichiometric $SrFeO_3$ obtained by equilibration at high oxygen pressures and below room temperatures possesses a metallic conductivity of about 500 S cm$^{-1}$ (if not further specified, we consider the following electrical conductivities measured at environmental conditions) [70]. At 500 °C, a maximum conductivity of 181 S cm$^{-1}$ was obtained [25]. Slight Mo incorporation decreases the conductivity since the reduction of $Fe^{4+}$ to $Fe^{3+}$ reduces the number of charge carriers and the hole mobility [24]. The conductivity values at 800 °C drop with increasing Mo fraction from 62 S cm$^{-1}$ for SF to 22 S cm$^{-1}$ for $SF_{1.5}M$ [25]. Values reported by other authors for $SF_{1.5}M$ at 800 °C are in the range of 7 to 21 S cm$^{-1}$ [30,31,35,39,40,44]. The conductivity of $SF_{1.5}M$ is much lower than that of SFM under similar conditions [60]. Here, the difference is a nearly random distribution of Fe and Mo at B-sites in $SF_{1.5}M$, while the B-site cation distribution in SFM is highly ordered. Ordering of $SF_{1+x}M$ continuously increases in the range of $0.5 > x \geq 0$ [28,36]. However, total B-site ordering requires a thermodynamic equilibrium during synthesis, which in ceramic materials is reached by long-term annealing at 1200 °C [69]. Note that in this work, we disregarded unusually high conductivity data in [12], which do not match with data measured under similar conditions and also with the dependence of the conductivity on the Mo fraction in $SF_{1+x}M$. A probable reason for the differences could be that the specified Fe fraction does not correspond to the present one.

The low-temperature part of the $SF_{1+x}M$ conductivities up to a transition temperature shifts to higher values with increasing Mo fraction. It is well described by the adiabatic small polaron hopping model [71]:

$$\rho = \rho_0 T \exp\left(\frac{E_a}{kT}\right), \tag{2}$$

where $E_a$ is the thermal activation energy. The decrease in conductivity at higher temperatures is related to a decreasing carrier concentration (holes in *p*-type materials) as the concentration of oxygen vacancies is increased due to the increasing loss of lattice oxygen. The generation of oxygen vacancies and electrons leads to a reduction in cations with high valence states to lower ones by electrons. Consequently, the effective charge carrier concentration will be reduced, causing a subsequent decrease in conductivity [72] and a corresponding decrease in the carrier density in this case [23,24]. This suggests that Mo-doped SFO has a stronger tolerance to reduction. For the same reason, the conductivity of $Sr_{1.9}Fe_{1.4}Ni_{0.1}Mo_{0.5}O_6$ drops below that of $Sr_2Fe_{1.5}Mo_{0.5}O_6$ when the temperature increases since the oxygen vacancies are easier formed in $Sr_{1.9}Fe_{1.4}Ni_{0.1}Mo_{0.5}O_6$ [35]. Note that the mobility of holes in $SF_{1+x}M$ decreases in response to a decrease in oxygen content in the lattice or an increase in Mo fraction. The latter effect is explained by the blocking role of $Mo^{6+}$ ions in the hole transfer via the Fe-O-Fe pathway [53]. Figure 2 illustrates the interplay of small adiabatic polaron hopping conduction at lower temperatures and loss of lattice oxygen at higher temperatures.

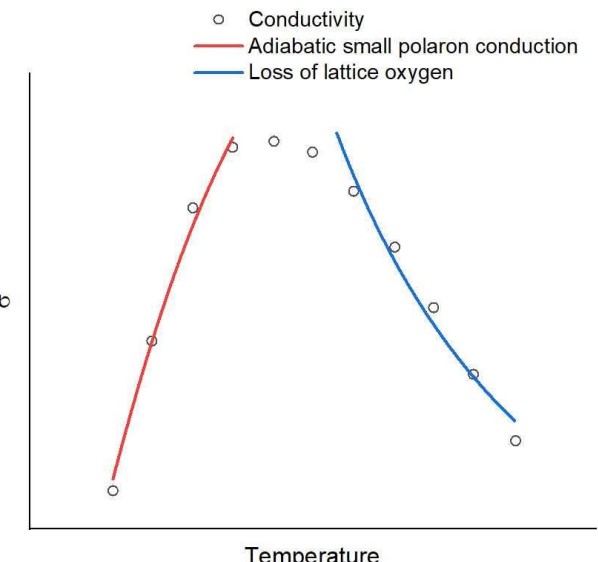

**Figure 2.** Schematic representation of the temperature dependence of conductivity in $SF_{1+x}M$ ceramics.

Similar behavior of the conductivity with a maximum in the temperature range 400–800 °C was also obtained for Ba substitution of Sr [37,44], Ni substitution of Fe [40], Co substitution of Fe [39,73] and Nd substitution of Mo [45] or of Fe [30]. Here, the low-temperature part was attributed to small adiabatic polaron hopping in [39,44,45,73,74]. Moreover, the temperature-activated behavior of hole mobility in $SF_{1+x}M$, $0.75 \leq x \leq 0.93$, was characteristic of the polaron conduction mechanism [53].

Monocrystalline, half-metallic SFM exhibits a low-temperature resistivity $\rho$ of $\rho_0 = 1.8 \times 10^{-6}$ $\Omega$ m increasing with temperature as:

$$\rho(T) = \rho_0 + R_\nu T^\nu, \tag{3}$$

with $\nu = 2$ near room temperature and the parameter $R_2 = 2.16 \cdot \times 10^{-11}$ $\Omega$ m K$^{-2}$ for single crystal SFM [75,76]. The $T^2$ dependence suggests that electron–electron scattering [76,77] or spin-wave scattering [76] dominates the resistivity. At higher temperatures up to the Curie temperature $T_C$, other charge scattering mechanisms appear in perovskite thin films, transforming Equation (3) into

$$\rho(T) = \rho_0 + R_2 T^2 + \sum_l R_l T^l, \tag{4}$$

where an $l = 2.5$ term represents the combination of electron–electron scattering, electron–phonon scattering and electron–magnon scattering [78,79]; an $l = 3$ term stands for the scattering with anomalous single magnons in half-metallic systems [80]; $l = 3.5$ with spin-waves at low temperatures [81,82]; an $l = 4.5$ with electron–magnon scattering derived in the double-exchange theory at low temperatures [83] but experimentally observed at mediate (200–350 K) temperatures [77] or possibly for spin-wave scattering [84]; and $l = 5$ with acoustic phonons [85]. Additionally, there is a term due to optical phonon scattering:

$$\rho(T) \propto \frac{\omega_s}{\sinh^2(\hbar\omega_s/2kT)}, \tag{5}$$

where $\omega_s$ is the average frequency of the softest optical mode, which is consistent with small polaron coherent motion involving relaxation [86]. Unfortunately, none of these models fit the experimental data of SFM thin films in the whole temperature range. However, the electron–electron scattering model is appropriate in a broader temperature range [87].

Above room temperature, the conductivity of SFM annealed in a vacuum is separated into three regions: (i) from 300 K up to the Curie temperature of about 405 K, where the

electrical resistivity increases with temperature with metallic behavior; (ii) above 405 K up to approximately 590 K, where conductivity decreases with temperature due to a B-site disorder induced weak Anderson localization of the electrical carrier; and (iii) from 590 K up to 900 K, where the material becomes metallic again [88]. Another report on the electrical resistivity of SFM indicates metallic conduction behavior below 150 °C, localization of the carriers in the temperature range of 150–550 °C, and reversion to metallic conduction behavior between 550 and 850 °C [60]. A similar conductivity behavior with a maximum of $\sigma$ at about 200 °C, far above the Curie temperature of ~60°C [89], was obtained for BFM, while CaFM shows solely metallic behavior in the whole temperature range of 50–850 °C. This correlates with the fraction of $Fe^{2+}$-$Mo^{6+}$ pairs discussed above [60]. The metallic resistivity of CaFM follows Equation (3) with $\rho_0 = 0.65$ m$\Omega$ cm, $\nu = 1.41$ and $R\nu = 0.157$ $\mu\Omega$ cm $T^{-\nu}$. This can be attributed to an electron scattering process operating from temperatures of the order of the Debye temperature up to the melting point. Here, the resistivity is proportional to the mean squared amplitude of atomic vibrations in the crystals varying with temperature according to a $T^{1.5}$ power law [90].

On the other hand, a revision of the reported conductivity behavior of SFM and BFM above $T_C$ [60,88] shows a convincing fit for the small adiabatic polaron hopping model [71] (cf. Equation (2)). The obtained $E_a$ values are 0.07–0.08 eV for SFM and about 0.13 eV for BFM. They are in the order of the values of other double perovskites compiled in Table 2.

**Table 2.** Activation energy of the adiabatic polaron hopping model of double perovskites measured under environmental conditions.

| Compound | Temperature Range, °C | $E_a$, eV | Ref. |
|---|---|---|---|
| LSCF | 100–500 | 0.10 | [91] |
| BSCF | 45–470 | 0.365 | [92] |
| SMgM | | 0.279 | |
| $S_{1.8}Sm_{0.2}MgM$ | | 0.195 | |
| $S_{1.6}Sm_{0.4}MgM$ | 450–850 | 0.113 | [93] |
| $S_{1.4}Sm_{0.6}MgM$ | | 0.080 | |
| $S_{1.2}Sm_{0.8}MgM$ | | 0.087 | |
| SFO | 400–500 | 0.204 | |
| $SF_{1.9}M$ | 400–600 | 0.221 | |
| $SF_{1.8}M$ | 400–600 | 0.164 | [25] |
| $SF_{1.6}M$ | 410–700 | 0.162 | |
| $SF_{1.5}M$ | 400–750 | 0.156 | |
| $SF_{1.5}M$ | 40–440 | 0.236 | |
| $SF_{1.0}C_{0.5}M$ | 40–530 | 0.193 | [39] |
| $SF_{0.5}C_{1.5}M$ | 40–530 | 0.137 | |
| $SF_{1.5}M$ | 300–450 | 0.067 | [45] |

SFO is *p*-type only under oxidizing conditions ($pO_2 > 10^{-6}$ atm) [53]. In reducing atmospheres, *p*-type conduction behavior is turned to *n*-type one. Thereby, the electron mobility is sufficiently lower than that of holes. It increases with rising Mo incorporation while the hole mobility decreases [23,24,53]. As shown above, the latter is explained by a blocking role of $Mo^{6+}$ ions in the hole transfer via the Fe-O-Fe pathway [53]. The increasing electron mobility, the *n*-type carrier, is a result of partial molybdenum reduction to $Mo^{5+}$ under reducing conditions which generates electrons. The $Mo^{5+}$/$Mo^{6+}$ ratio shows a peak in $SF_{1.5}M$ [22]. Partial substitution of Fe at the B-site by an ion of a different radius affects the conductivity. In air, the highest conductivity is obtained as a result of substitution

when the tolerance factor $t$ adopts the ideal value $t = 1$ [26]. This was also observed by the substitution of A-site divalent ions by trivalent $Nd^{3+}$ and $Sm^{3+}$ in BSCF [92].

$SF_{1.33}M$ shows a metallic behavior from 370 to 830 °C in 3% $H_2O/H_2$. At 800 °C, the conductivity amounts to about 16 S cm$^{-1}$ [34]. A fit to Equation (3) yields $\rho(0) = 18.8 \cdot m\Omega \cdot cm$, $\nu = 1.54$ and $R_\nu = 1.482 \cdot \mu Q$ cm $T^{-\nu}$. Moreover, in these cases, the temperature dependence of conductivity originates in the high-temperature electron scattering process exhibiting a $T^{1.5}$ power law [90]. Table 3 lists the fits of experimental conductivity behavior of $SF_{1+x}M$ samples possessing metallic conductivity to Equation (3).

**Table 3.** Parameters of metallic conductivity of $SF_{1+x}M$ measured under reducing conditions.

| Compound | $\rho(0)$, m$\Omega$ cm | $\nu$ | $R_\nu$, $\mu\Omega$ cm $T^{-\nu}$ | Ref. |
|---|---|---|---|---|
| $SF_{1.33}M$ | 18.8 | 1.54 | 1.482 | [34] |
| SFM | 1.85 | 2.07 | 0.0013 | [93] |
| SFM | 3.01 | 1.75 | 0.014 | [94] |

At 800 °C and in reducing atmospheres, $SF_{1.5}M$ conductivities of 8 to 40 S cm$^{-1}$ were obtained [22,29–31,41,56,95]. A reason for this data scatter is the strong dependence of the conductivity on the Fe-fraction. For example, it varies from 16 S cm$^{-1}$ for $SF_{1.4}M$ to 75 S cm$^{-1}$ for $SF_{1.6}M$ [22]. We attributed this increase to an increase for $x < 0.5$ in the B-site order of $SF_{1+x}M$ [28,36]. The value of $\sigma$ increases in strontium deficient $Sr_{1.9}Fe_{1.4}Ni_{0.1}Mo_{0.5}O_{6-\delta}$ up to about 29 S cm$^{-1}$ benefiting from the high conductivity of a cubic Fe-Ni metal alloy formed under reducing conditions [35]. Under reducing conditions, the temperature range of small adiabatic polaron hopping conduction widens to higher temperatures. Table 4 compiles the activation energies of $SF_{1+x}M$ ceramics obtained from experimental data, which are subjected to this conduction process. $E_a$ values are of the order of 0.1 eV with a large scatter attributed to differences in sample synthesis and post-treatment.

**Table 4.** Activation energy of the adiabatic polaron model of $SF_{1.5}M$-based ceramics under reducing conditions.

| Compound | Temperature Range, °C | $E_a$, eV | Ref. |
|---|---|---|---|
| $SF_{1.5}M$ | 475–800 | 0.167 | [29] |
| $SF1.5M$ | 475–800 | 0.292 | [35] |
| $S_{1.9}F_{1.4}Ni_{0.1}M$ | 400–800 | 0.240 | [35] |
| $SF_{1.5}M$ | 200–800 | 0.225 | [56] |
| $SF_{1.5}M$ | 400–750 | 0.15 | [32] |
| $SF_{1.5}M$ ($SF_{1.4}N_{0.1}M$) | 300–800 | 0.115 (0.1) | [30] |
| $SFM_{0.8}N_{0.2}$ | 600–800 | 0.281 | [96] |
| $SFM_{0.65}Ni_{0.35}$ | 250–700 | 0.05 | [97] |

Over the temperature range of 50–850 °C, the electrical resistivity of $A_2FeMoO_{6-\delta}$ in $H_2$ shows a clear increase by several orders of magnitude in the order: CaFM < SFM < BFM [60]. More detailed data were not provided. The conductivity of SFM sintered in 5 % $H_2/Ar$ at 1000 °C for 5 h and measured in $H_2$ exhibits a small adiabatic polaron hopping conduction [94].

Under reducing conditions, the conductivity decreases with increasing ion radius of the ion partially substituting Fe in SF. It is maximum in the case of $Mo^{6+}$ substitution and decreases with the ionic radii of the dopant from 4.5 S cm$^{-1}$ for Mo-doping to 0.4 S cm$^{-1}$ for Zr-doping at 800 °C [26].

## 3. A- and B-Site Substitution

The partial substitution of A and/or B site elements is an effective way to improve the performance of perovskite-type materials. The additional introduction of alkali earth metal

elements on the A-site and transition-metal elements on the B-site affects the cation valence and oxygen-vacancy concentration and thus improves the electronic or ionic conductivity of the material as well as its catalytic properties.

### 3.1. A-Site Substitution

With a decrease in the A-site cation size in $A_2FeMoO_6$ (A = Ca, Sr, Ba), the $Fe^{3+}/Mo^{5+}$ electronic configuration increases resulting in an increase in the electrical conductivity of the material [60]. The thermal expansion coefficient does not change sufficiently with A-site substitution. The best performance of SOFCs with an $A_2FeMoO_6$ anode, a power density of 0.584 W cm$^{-2}$, was obtained for unsubstituted SFM (cf. Table 5) [60]. The change in the $Fe^{2+}/Fe^{3+}$ and $Mo^{5+}/Mo^{6+}$ ratios in A-site substituted SFM (cf. Section 2.3) plays an important role in the conduction and electrochemical performance [37,44,60]. When $x = 0.2$, the conductivity of the $SBa_{0.2}F_{1.5}M$ sample reaches the maximum value of 21.7 S cm$^{-1}$ at 550–600 °C where the dominant conductivity mechanism changes from small adiabatic polaron hopping to carrier diminution caused by oxygen lattice loss (cf. Section 2.5) [37]. This value is higher than that of the $SF_{1.5}M$ ($\sigma$ = 15.9 S cm$^{-1}$) sample [40]. $SBa_{0.2}F_{1.5}M$ shows the lowest cathode polarization resistance and provides at 800 °C with Ni-YSZ anode and a YSZ electrolyte a maximum power density of 1.63 W cm$^{-2}$ at 800 °C (cf. Table 5) [37].

**Table 5.** Performance of $S_2F_{1+x}M_{1-x}$-based SOFCs operated with hydrogen fuel.

| Configuration Anode-Electrolyte-Cathode | $d_{el}$, μm | T, °C | $P_{max}$, W/cm$^2$ | Ref. |
|---|---|---|---|---|
| SFM/LSGM/SDC/SmBC$_2$ | 300 | 850 | 0.831 | [60] |
| | | 800 | 0.584 | |
| | | 750 | 0,412 | |
| SFM/LSGM/SDC/SmBC$_2$ | 300 | 850 | 0.735 [1] | [60] |
| | | 800 | 0.476 [1] | |
| | | 750 | 0.183 [1] | |
| BFM/LSGM/SDC/SmBC$_2$ | 300 | 850 | 0.561 | [60] |
| | | 800 | 0.338 | |
| | | 750 | 0.206 | |
| CaFM/LSGM/SDC/SmBC$_2$ | 300 | 850 | 0.186 | [60] |
| SFM/LSGM/BSCF | 300 | 850 | 0.864 | [61] |
| | | 800 | 0.603 | |
| | | 750 | 0.436 | |
| SFN$_{0.8}$M$_{0.2}$/LSGM/PBCO | 200 | 800 | 0.520 | [96] |
| | | 700 | 0.375 | |
| | | 600 | 0.130 | |
| | | 550 | 0.061 | |
| SF$_{1.33}$M/LGSM/LSCF | 300 | 800 | 0.547 | [34] |
| | | 750 | 0.392 | |
| | | 700 | 0.268 | |
| SF$_{1.33}$M/LGSM/LSCF | 300 | 800 | 0.472 [1] | [34] |
| SFM/LSGM/BSCF | 300 | 850 | 0.864 | [61] |
| | | 800 | 0.603 | |
| | | 750 | 0.436 | |

**Table 5.** *Cont.*

| Configuration Anode-Electrolyte-Cathode | $d_{el}$, μm | $T$, °C | $P_{max}$, W/cm² | Ref. |
|---|---|---|---|---|
| $SF_{1.33}M$/LGSM/LSCF | 300 | 800 | 0.472 [1] | [34] |
| $SF_{1.33}M$/LGSM/LSCF | 265 | 850 | 0.532 | [98] |
| | | 800 | 0.340 | |
| | | 750 | 0.200 | |
| $SF_{1.33}M$/LGSM/LSCF $SF_{1.5}M$/LGSM/LSCF | NA | 800 | 0.588 0.474 | [29] |
| $SF_{1.33}Mo_{0.66}$/LGSM/LSCF $SF_{1.5}M$/LGSM/LSCF | NA | 800 | 0.473 [1] 0.432 [1] | [29] |
| Ni-$SF_{1.5}M$/LGSM/LSCF | 300 | 800 | 1134 | [99] |
| $SF_{1.5}M$-GDC/CeGdO/BCFN | 50 | 700 | 0.188 | [100] |
| | | 650 | 0.100 | |
| | | 600 | 0.039 | |
| $S_{1.9}F_{1.4}Ni_{0.1}M$/LSGM/CLO/LSCF | NA | 850 | 1.160 | [35] |
| | | 800 | 0.968 | |
| | | 750 | 0.730 | |
| $SF_{1.5}M$-GDC/GDC/GDC-LSCF | 35 | 700 | 0.22 | [95] |
| | | 600 | 0.14 | |
| LSMn/YSZ-$S_2FM$/YSZ/YSZ-LSFSc | 24 | 800 | 0.462 | [52] |
| | | 750 | 0.324 | |
| $SF_{1.4}M$/LSGM/BSCF $SF_{1.5}M$/LSGM/BSCF $SF_{1.6}M$/LSGM/BSCF | 300 | 800 | 0.514 0.508 0.482 | [22] |
| $SF_{1.4}M$/LSGM/BSCF $SF_{1.5}M$/LSGM/BSCF $SF_{1.6}M$/LSGM/BSCF | 300 | 750 | 0.387 0.385 0.380 | [22] |
| $SF_{1.3}Co_{0.3}M_{0.4}$/LSGM/LSCF | 170 | 850 | 1.09 | [74] |
| | | 800 | 0.81 | |
| | | 750 | 0.50 | |
| | | 700 | 0.30 | |
| $SF_{1.3}Co_{0.3}M_{0.4}$/LSGM/LSCF | 170 | 850 | 0.981 [2] | [74] |
| | | 800 | 0.808 [2] | |
| | | 750 | 0.604 [2] | |
| Ni-LDC/LDC/LSGM/$SF_{1.5}M$ | 300 | 850 | 0.613 | [101] |
| | | 800 | 0.468 | |
| | | 750 | 0.349 | |
| Ni-LDC/LDC/LSGM/$SF_{1.5}M$ | 265 | 850 | 0.613 | [98] |
| | | 800 | 0.468 | |
| | | 750 | 0.349 | |
| Ni-YSZ/YSZ/SDC/$S_{1.8}B_{0.2}F_{1.5}M$ | 700 | 800 | 1.63 | [37] |
| | | 750 | 1.3 | |
| | | 700 | 0.87 | |
| | | 650 | 0.41 | |

**Table 5.** *Cont.*

| Configuration Anode-Electrolyte-Cathode | $d_{el}$, μm | $T$, °C | $P_{max}$, W/cm² | Ref. |
|---|---|---|---|---|
| Ni-YSZ/YSZ/SDC/SF$_{1.5}$Mo$_{0.4}$N$_{0.1}$ | 400 | 800 | 1.102 | [45] |
| | | 750 | 0.920 | |
| | | 700 | 0.671 | |
| | | 650 | 0.421 | |
| Ni-YSZ/YSZ/SDC/S$_{1.8}$B$_{0.2}$F$_{1.5}$M Ni-YSZ/YSZ/SDC/S$_{1.6}$B$_{0.4}$F$_{1.5}$M Ni-YSZ/YSZ/SDC/S$_{1.4}$B$_{0.6}$F$_{1.5}$M | 400 | 800 | 1.06 1.26 0.94 | [44] |
| Ni-YSZ/YSZ/SDC/SF$_{1.4}$Ni$_{0.1}$M | 10 | 800 | 1.77 | [40] |
| | | 750 | 1.21 | |
| | | 700 | 0.79 | |
| | | 650 | 0.33 | |
| Ni-YSZ/LSGM/SDC/SF$_{1.5}$M Ni-YSZ/LSGM/SDC/SF$_{1.45}$Sc$_{0.05}$M | 400 | 800 | 0.91 1.23 | [42] |
| Ni-ScSZ/ScSZ/SDC/SF$_{1.4}$Co$_{0.1}$M Ni-ScSZ/ScSZ/SDC/S$_{1.95}$F$_{1.4}$Co$_{0.1}$M Ni-ScSZ/ScSZ/SDC/S$_{1.9}$F$_{1.4}$Co$_{0.1}$M | 11 | 800 | 0.88 1.16 0.96 | [73] |
| SF$_{1.5}$M/LSGM/SF$_{1.5}$M | 265 | 900 | 0.835 | [12] |
| SF$_{1.5}$M/LSGM/SF$_{1.5}$M | 265 | 900 | 0.835 | [98] |
| SF$_{1.5}$M/LSGM/SF$_{1.5}$M | 243 | 800 | 0.531 | [30] |
| | | 750 | 0.365 | |
| | | 700 | 0.244 | |
| | | 650 | 0.124 | |
| SF$_{1.4}$N$_{0.1}$M/LSGM/SF$_{1.5}$M | 236 | 800 | 0.374 | [30] |
| | | 750 | 0.228 | |
| | | 700 | 0.156 | |
| | | 650 | 0.092 | |
| S$_{1.4}$Ca$_{0.6}$F$_{1.5}$M/LSGM/S$_{1.4}$Ca$_{0.6}$F$_{1.5}$M | 35 | 800 | 1.050 | [38] |
| | | 750 | 0.880 | |
| | | 700 | 0.660 | |
| | | 600 | 0.410 | |
| SF$_{1.4}$Ni$_{0.1}$M/LSGM/SF$_{1.4}$Ni$_{0.1}$Mo$_{0.5}$ | 310 | 800 | 0.530 [3] | [41] |
| | | 750 | 0.380 | |
| | | 700 | 0.258 | |
| | | 650 | 0.164 | |
| SF$_{1.5}$M/LSGM/SF$_{1.5}$M$_{0.2}$Sn$_{0.3}$ | 400 | 800 | 0.618 | [43] |
| | | 750 | 0.431 | |
| | | 700 | 0.262 | |

[1] H$_2$S/H$_2$, [2] syngas, [3] anode decomposes.

Partial substitution of Sr$^{2+}$ by Ca$^{2+}$ changes the cathode polarization resistance in air, showing the lowest value at $x = 0.4$ and decreasing the thermal expansion coefficient toward a better thermal match with the electrolyte [44]. On the other hand, the substitution effectively enhances the anode catalytic activities, thereby reducing the anode polarization resistance, especially at lower temperatures [38].

In symmetrical SOFCs, the anode polarization resistances of SCa$_x$FM, $x$ = 0, 0.2, 0.4, 0.6 and electrodes are significantly higher than that of the cathode. It decreases with an increasing value of $x$, while for the cathode, a minimum at compositions $x$ = 0.2–0.4 occurs [38,44]. The lowest anode polarization resistance of SC$_{0.6}$M is coupled with slightly higher cathode polarization resistances than those of SF$_{0.4}$M. SOFC measurements showed the (up until now) highest power densities for symmetrical SC$_{0.6}$M electrodes and an LSGM electrolyte, 1.06 W cm$^{-2}$ at 800 °C (cf. Table 5) [38]. The thermal expansion coefficients were found to decrease when increasing the Ca content from 16.33 × 10$^{-6}$ K$^{-1}$ to 15.08 × 10$^{-6}$ K$^{-1}$ when the Ca-substitution increases from 0 to 0.6 [44]. This leads to a better match with LSGM values of 11.5 × 10$^{-6}$ K$^{-1}$ for a wide range of temperatures (from room temperature to 1000 °C) [16]. In summary, the partial substitution of Sr$^{2+}$ by Ca$^{2+}$ improves the performance of the cathode of IT-SOFCs (cf. Table 5).

### 3.2. B-Site Substitution

Iron and cobalt oxides are well known for their excellent catalytic activity. On the other hand, Sr$_2$CoMoO$_{6-\delta}$ and Sr$_2$NiMoO$_{6-\delta}$ exhibit low resistance to carbonation in the intermediate temperature range of 600–800 °C but are unstable under a reducing atmosphere above 800 °C, decomposing partially into Sr$_3$MoO$_6$ and Co or Ni metals, respectively [17]. Recently, Sr$_2$Fe$_{1.5-x}$Co$_x$Mo$_{0.5}$O$_{6-\delta}$ (SF$_{1+x}$C$_x$M) [39] and Sr$_2$Fe$_{1.5-x}$Ni$_x$Mo$_{0.5}$O$_{6-\delta}$ (SF$_{1+x}$N$_x$M) cathodes [40] have been reported. Both studies demonstrated that partial B-site substitution with transition-metal elements enhances the conductivity and electrochemical performance.

Fe$^{2+}$/Fe$^{3+}$ substitution by Co$^{2+}$/Co$^{3+}$ in SF$_{1.5-x}$Co$_x$M improves the conductivity, increases oxygen vacancy concentration and, consequently, enhances the surface exchange kinetics without deteriorating the structural stability. Unfortunately, it also significantly increases the thermal expansion coefficient from 15.8 to 19.8 × 10$^{-6}$ K$^{-1}$[39]. The large TEC of the Co-containing perovskite (~20 × 10$^{-6}$ K$^{-1}$) is due to the formation of oxygen vacancies and spin-state transitions associated with Co$^{3+}$ [102].The temperature dependence of $\sigma$ shows the typical behavior where the dominant conductivity mechanism changes from small adiabatic polaron hopping to carrier diminution caused by oxygen lattice loss (cf. Section 2.5). For $x$ = 0, the maximum conductivity amounts to about 27 S cm$^{-1}$ at 645 °C, $x$ = 0.5 to about 59 S cm$^{-1}$ at 535°C, while for $x$ = 1, it increases to about 118 S cm$^{-1}$ at 450 °C [39]. The improvement of the surface exchange coefficient by about two orders of magnitude at 750 °C from 2.55 × 10$^{-5}$ cm s$^{-1}$ for $x$ = 0 to 2.20 × 10$^{-3}$ cm s$^{-1}$ for $x$ = 1.0 was similar to LaCoO$_3$ attributed to electron occupation of the crystal field d state near Fermi level and with the buildup of surface charge so as to enhance the electron transfer between a surface cation and a potentially catalyzed species [6]. Similar to SF$_{1.5}$M [103], the electrochemical performance of S$_x$F$_{1.4}$Co$_{0.1}$M can be further improved by Sr deficiency. Sr-deficiency shrinks the cubic crystal lattice. The TEC value first decreases from 16.40 × 10$^{-6}$ K$^{-1}$ for $x$ = 0 to the minimum of 15.62 × 10$^{-6}$ K$^{-1}$ at $x$ = 1.950 and then increases to 16.16 × 10$^{-6}$ K$^{-1}$ for $x$ = 1.900. The processes of Fe$^{3+}$ + Mo$^{5+}$ ⇔ Fe$^{2+}$ + Mo$^{6+}$ and Co$^{3+}$ + Mo$^{5+}$ ⇔ Co$^{2+}$ + Mo$^{6+}$ equilibriums were not obviously affected by increasing the number of Sr-site deficiencies. On the other hand, the conductivity increases slightly up to a maximum of ~27 S cm$^{-1}$ for S$_{1.95}$F$_{1.4}$Co$_{0.1}$M at 500 °C where the dominant conductivity mechanism changes from small adiabatic polaron hopping to carrier diminution caused by oxygen lattice loss (cf. Section 2.5) and when decreases with increasing $x$ again to S$_{1.9}$F$_{1.4}$Co$_{0.1}$M. An anode-supported single cell with a thin scandia-stabilized yttria (ScSZ) electrolyte film (11 µm) on a Ni-ScSZ anode and a CSO interlayer to the S$_{1.95}$F$_{1.4}$Co$_{0.1}$M cathode showed a $P_{max}$ of 1.16 W cm$^{-2}$ at 800 °C (cf. Table 5) [73].

Active cobalt metal nanoparticles are in situ exsolved in reducing atmosphere from the parent SF$_{1.3}$Co$_{0.2}$M. The maximum electrical conductivity in 5 %H$_2$/N$_2$ at 600 °C amounts to 13.9 S cm$^{-1}$. SOFCs achieved maximum power densities at 850 °C of 1.09, 0.981 and 0.29 W cm$^{-2}$ when hydrogen, syngas and methane were used as fuel, respectively (cf. Tables 5 and 6). Continuous operation in hydrogen for 115 hs, in syngas for 190 hs and in methane for 300 hs did not show degradation [74].Substitution of Fe$^{2+}$/Fe$^{3+}$ by Co$^{3+}$

in $SFC_xM$ decreases B-site ordering and increases the conductivity monotonically up to a metallic behavior for $x \geq 0.15$ with $\sigma \geq 135$ S cm$^{-1}$. Thereby, the Co ion is in the trivalent, high-spin state [104].

**Table 6.** Performance of $SF_{1+x}M$-based SOFCs operated with hydrocarbons.

| Configuration Anode-Electrolyte-Cathode | Fuel | $d_{el}$, μm | $T$, °C | $P_{max}$, W/cm$^2$ | Ref. |
|---|---|---|---|---|---|
| SFM/LSGM/BSCF | CH$_4$ | 300 | 850 | 0.605 | [61] |
| | | | 800 | 0.429 | |
| SF$_{1.33}$M/LGSM/LSCF | CH$_4$ | 300 | 800 | 0.13 | [34] |
| SF$_{1.33}$M/LGSM/LSCF SF$_{1.5}$M/LGSM/LSCF | CH$_4$ | NA | 800 | 0.079 0.041 | [29] |
| SF$_{1.5}$M/LSGM/SF$_{1.5}$M | CH$_4$ | 265 | 900 | 0.23 | [99] |
| SF$_{1.5}$M/LSGM/SF$_{1.5}$M | CH$_4$ | 400 | 900 | 0.250 | [31] |
| | | | 850 | 0.125 | |
| Ni-SF$_{1.5}$M/LGSM/LSCF | CH$_4$ | 300 | 800 | 0.663 | [99] |
| SF$_{1.3}$Co$_{0.3}$M/LSGM/LSCF | CH$_4$ | 170 | 850 | 0.290 | [74] |
| | | | 750 | 0.151 | |
| | | | 700 | 0.057 | |
| SFM-YSZ/YSZ/YSZ-LSFSc | C$_3$H$_8$ | 24 | 800 | 0.331 | [52] |
| | | | 750 | 0.173 | |
| SF$_{1.4}$M/LSGM/BSCF SF$_{1.5}$M/LSGM/BSCF SF$_{1.6}$M/LSGM/BSCF | CH$_3$OH | 300 | 800 | 0.415 0.395 0.382 | [22] |
| SF$_{1.4}$M/LSGM/BSCF SF$_{1.5}$M/LSGM/BSCF SF$_{1.6}$M/LSGM/BSCF | CH$_3$OH | 300 | 750 | 0.341 0.297 0.205 | [22] |

Ni$^{2+}$ substitutes Fe$^{2+}$ because the ionic radius of Ni$^{2+}$(69 pm] [21]) is smaller than that of Fe$^{2+}$ (78 pm [21]) while larger than Fe$^{3+}$ (64.5 pm [21]). It promotes oxygen vacancy formation on the SF$_{1.5}$M surface, which in turn improves the electrochemical performance toward fuel oxidation [105]. The conductivity reaches 60 S cm$^{-1}$ at 450 °C in air where the dominant conductivity mechanism changes from small adiabatic polaron hopping to carrier diminution caused by oxygen lattice loss (cf. Section 2.5). At the same composition, the lowest cathode polarization occurs, which is only 50 % of the SF$_{1.5}$M value. The thermal expansion coefficient increases from $15.6 \times 10^{-6}$ K$^{-1}$ to $18.1 \times 10^{-6}$ K$^{-1}$ with $x$ varying from 0.05 to 0.4. This increases the thermal mismatch with the electrolyte. The polarization resistance of an SF$_{1.4}$Ni$_{0.1}$M cathode was approximately 50% of that of the SF$_{1.5}$M cathode. The maximal power density amounts to 1.77 W cm$^{-2}$ at 800 °C [40], exceeding the improvement achieved by A-site substitution (cf. Table 5). SF$_{1.5-x}$Ni$_x$M, after reduction at 750 °C, possesses a single-phase perovskite structure up to $x = 0.1$. For higher Ni content ($x = 0.2$ and 0.3), a NiO impurity phase appears, while for $x = 0.4$, a Sr$_3$FeMoO$_{6.5}$ phase gives evidence of SF$_{1.5-x}$Ni$_x$M decomposition. At 800 °C, Sr$_3$FeMoO$_{6.5}$ was also formed in the x = 0.1 sample. The conductivity of SF$_{1.4}$Ni$_{0.1}$M in H$_2$ monotonously increases in the temperature range 600 to 800 °C reaching a maximum value of 20.6 S cm$^{-1}$. The minimum anode polarization resistance occurs at the same composition. SOFCs show stable performance for 15 h providing a maximum power density of 0.380 W cm$^{-2}$ at 750 °C (cf. Table 5) [41]. Thus, similar to a Ni solubility limit of 13 to 18 % in the La$_{0.8}$Sr$_{0.2}$Cr$_{1-x}$Ni$x$O$_{3-\delta}$ lattice [106], a solubility of ~20% may be considered for SF$_{1.5}$M [41].

The introduction of a slight Sr-site deficiency favors the metal precipitation in reducing atmospheres. By this way a Ni-Fe alloy phase was created in S$_{1.9}$F$_{1.4}$Ni$_{0.1}$M which improved

the catalytic activity, the redox stability and the SOFC maximum power density reaching $0.968 \text{ W cm}^{-2}$ at 800 °C with $H_2$ as a fuel and $0.277 \text{ W cm}^{-2}$ at 800 °C with $C_3H_8$ as a fuel (cf. Tables 5 and 6) [35]. By dispersing a small amount of Ni (~2 wt%) on the $SF_{1.5}M$ ceramic anode, the performance of SOFCs with LSGM as electrolyte and LSCF as a cathode has been significantly improved both in $H_2$ and $CH_4$ as the fuel and ambient air as the oxidant (cf. Tables 5 and 6). It is very stable when operating with $CH_4$ fuel and provides the highest maximal power density, $0.663 \text{ W cm}^{-2}$, reported for $CH_4$ fuel at 800 °C (Table 6). This way, the carbon formation on the Ni surface can be suppressed by controlling the dispersion and loading of Ni on the $SF_{1.5}M$ anodes [99].

Nb substitutes are either Mo or Fe since the ionic radii of $Nb^{4+}$ (68 pm [21]) and $Nb^{5+}$ (64 pm [21]) are only slightly larger than that of $Mo^{5+}$ (61 pm [21]) and $Mo^{6+}$ (59 pm [21]) and high-spin $Fe^{3+}$ (64.5 pm [21]), expanding the lattice [21]. Under reducing conditions, $Fe^{3+}$ is easily reduced. Here, high-spin $Fe^{3+}$ (64.5 pm) substitution by $Nb^{5+}$ increases the stability of $SF_{1.5}M$. On the other hand, the crystal lattice expands even though the cation size of $Nb^{5+}$ is smaller than that of $Fe^{3+}$. This was attributed to the reduction of $Fe^{3+}$ ions in air $Fe^{3+}$ to $Fe^{2+}$ with a larger ion radius (76 pm) in order to achieve electrical neutrality. An increase in the $Fe^{2+}$ fraction from 36.4% in $SF_{1.5}M$ to 49.5% in $SF_{1.4}N_{0.1}M$ was proved by XRD [30]. Although this is not direct evidence for $Fe^{3+}$ substitution by $Nb^{4+}$, we accept this explanation here. Note that for Mo substitution, Nb exhibits lower mixed valences (+4/+5) than that of Mo (+5/+6). When Mo is substituted by Nb, the value of δ increases, demonstrating an increase in the oxygen vacancy concentration. Unfortunately, the thermal expansion coefficient values of $SF_{1.5}M_{0.5−x}N_x$ powders increase from $15.6 \times 10^{-6} \text{ K}^{-1}$ to $16.7 \times 10^{-6} \text{ K}^{-1}$ when x ranges from 0.05 to 0.20, increasing the thermal mismatch with the electrolyte. The maximum conductivity is obtained for $SF_{1.5}M_{0.4}N_{0.1}$, amounting to $31 \text{ S cm}^{-1}$ at 550 °C where the dominant conductivity mechanism changes from small adiabatic polaron hopping to carrier diminution caused by oxygen lattice loss (cf. Section 2.5). The same composition exhibits the lowest polarization resistance. The power density of a Ni-YSZ anode-supported SOFC consisting of an $SF_{1.5}M_{0.5−x}N_x$ cathode reaches $1.1 \text{ W cm}^{-2}$ at 800 °C (cf. Table 5). No obvious performance degradation is observed over 15 h at 750 °C with wet $H_2$ (3 % $H_2O$) as fuel and ambient air as the oxidant [45]. The electrical conductivities of $SF_{1.4}N_{0.1}M$ increase gradually with temperature up to the maximum value at 600 °C, where the dominant conductivity mechanism changes from small adiabatic polaron hopping to carrier diminution caused by oxygen lattice loss (cf. Section 2.5), and then decreases. At 600 °C, the substitution of 6.7 % $Fe^{2+}/Fe^{3+}$ by $Nb^{5+}$ increases the conductivity from 17.62 to $27.61 \text{ S cm}^{-1}$ in air and from 11.11 to $15.86 \text{ S cm}^{-1}$ in 3 %$H_2O/H_2$. Furthermore, $SFe_{1.4}N_{0.1}M$ exhibits a lower polarization resistance under both oxidizing and reducing atmospheres. A symmetrical SOFC showed excellent redox stability and provided at 800 °C a maximum power density of $0.531 \text{ W cm}^{-2}$ (cf. Table 5). $SFe_{1.4}N_{0.1}M$-based SOFCs maintained a stable output over ten redox cycles at 750 °C, while for $SF1.5M$ one, the output declined obviously after seven cycles [30].

An $SFM_{0.8}N_{0.2}$ anode shows outstanding performances with high resistance against carbon build-up and redox cycling in hydrocarbon fuels. At 800 °C, it shows electrical conductivity of $5.3 \text{ S cm}^{-1}$ in 5% $H_2$ and provides maximum power densities of 0.520 and $0.380 \text{ W cm}^{-2}$ when using $H_2$ and $CH_4$ as the fuel, respectively (cf. Tables 5 and 6). After six redox cycles, there is no degradation observed in the cell voltage. Under different current loads, the SOFC showed impressive performance stability [96].

$Fe^{2+}$ (78 pm [21])/$Fe^{3+}$ (64.5 pm [21]) substitution by $Sc^{3+}$ (74.5 pm [21]) expands the lattice. Samples of $SF_{1.5−x}Sc_xM$ with x between 0.05 and 0.1 display the same initial cubic perovskite single-phase without no impurity. On the other hand, a $Sc_2O_3$ phase is observed when x is further increased up to 0.20. The substitution practically does not change the thermal expansion coefficient. The maximum conductivities of $SF_{1.5}M$ and $SF_{1.45}Sc_{0.05}M$ reach $17 \text{ S cm}^{-1}$ and $27 \text{ S cm}^{-1}$ in the temperature regions of 500–550 °C and 600–650 °C, respectively. The conductivity maximum is caused by a change in the conductivity mechanism from small adiabatic polaron hopping to carrier diminution caused

by oxygen lattice loss (cf. Section 2.5). Enhancing the conductivity and oxygen vacancy by $Sc^{3+}$ at the B-site of SFM promotes the reduction reaction of the adsorbed oxygen atom into an oxygen ion. Here, the dissociation of adsorbed molecular oxygen becomes the rate-limiting step. Ni-YSZ anode supported SOFCs with LSGM as an electrolyte, and an SDC layer at the cathode interface provides 800 °C maximum power densities of 1.23 and 0.91 W cm$^{-2}$ for $SF_{1.45}Sc_{0.05}M$ and $SF_{1.5}M$, respectively. No obvious degradation of the SOFC performance occurred in a 100 h test at 750 °C [42].

$Sn^{4+}$ favors the cubic structure of $SrSnO_3$. During the fabrication of $Sr_2Fe_{1.5}Mo_{0.5-x}Sn_xO_{6-\delta}$, tin adopts both the 2+ and 4+ valence. The substitution of $Mo^{5+}/Mo^{6+}$ by $Sn^{2+}/Sn^{4+}$ increases the TEC in the range of 20 °C to 900 °C from $16.26 \times 10^{-6}$ to $20.32 \times 10^{-6}$ K$^{-1}$ when $x$ increases from 0 to 0.5 increasing the thermal mismatch with the electrolyte. An electrolyte-supported single cell, consisting of $SF_{1.5}M$ anode, LSGM electrolyte and $SF_{1.5}M_{0.2}Sn_{0.3}$ cathode, provided a power density of 0.618Wcm$^{-2}$ at 800 °C which is comparable with the SOFC performance one of the other B-site substitutions but higher than the one of a symmetrical $SF_{1.5}M$-based cell (cf. [101] and Table 5). No obvious SOFC degradation was observed within the 200 h operation at 800 °C [43].

## 4. Performance of $Sr_2Fe_{1+x}Mo_{1-x}O_{6-\delta}$-Based SOFCs

### 4.1. Polarization Resistance

Polarization resistance ($R_p$) is the resistance at the interface between the electrodes and the electrolyte. It is caused by the electron transfer resistance and the gas diffusion resistance at the electrodes and reduces the open-circuit voltage of the SOFC. At high temperatures, the oxygen ion transport process is significantly accelerated. Consequently, reducing the SOFC operating temperature decreases the electrode kinetics and results in large interfacial polarization resistances. This effect is most pronounced for the oxygen reduction reaction at the cathode [6]. Thus, $R_p$ is an effective parameter to probe the catalytic activity of SOFC cathode for the oxygen reduction reaction (ORR). The relatively low operation temperature of IT-SOFCs decreases the electrode kinetics and the catalytic activity of the ORR and inhibits the diffusion of oxygen [107]. In this case, the performance of the cathode becomes a limiting factor of IT-SOFC performance. Here, $R_p$ is affected by the generation of oxygen vacancies at the electrode/electrolyte interface region induced by polarization [108], the increased concentration of oxygen vacancies by increasing the Fe/Mo ratio in $SF_{1+x}M$ enhancing catalytic activity, and the sintering temperature [101]. In order to enhance the catalytic activity of the ORR, significant effort has been devoted to studying new cathode materials by metal element substitution or doping while keeping the cubic crystal structure of the $SF_{1+x}M$ materials.

$R_p$ generally varies with oxygen partial pressure. It accounts for a number of physico-chemical reactions: (i) molecular oxygen adsorption on the catalyst surface, (ii) dissociation of adsorbed molecular oxygen, (iii) ionization of adsorbed oxygen atom, (iv) migration of oxygen ions to the triple phase boundary (TPB), (v) ionization of oxygen ions at the TPB and (iv) charge transfer at the TPB. Since $R_p$ and oxygen partial pressure are connected by a power law specific for each of the physicochemical reactions, studying the effect of oxygen partial pressure on $R_p$ allows analyzing the total reaction rate-limiting physicochemical reaction [42].

Electrochemical impedance spectroscopy (EIS) is an important diagnostic tool for the evaluation of polarization resistance. However, the impedance spectrum of a SOFC convolutes anode and cathode behavior, even when using reference electrodes. A simple equivalent circuit of a SOFC is a series connection of two constant phase elements *CPE* in parallel with a resistance *R* and an ohmic resistance $R_{ohm}$ overall ohmic resistance including the electrolyte resistance, electrode ohmic resistance and lead resistance, i.e., $(R_{HF} || CPE_{HF})$-$R_{ohm}$-$(R_{LF} || CPE_{LF})$ [101,109]. The resistance at the high-frequency $R_{HF}$ is probably associated with the charge-transfer processes, which include the ion-transfer process occurring at the electrode/electrolyte interfaces and the electron-transfer process accompanying the oxygen reduction reaction. The low-frequency part $(R_{LF} || CPE_{LF})$ can be

attributed to the diffusion processes, which include the adsorption–desorption of oxygen, oxygen diffusion at the gas–cathode interface and surface diffusion of intermediate oxygen species. The difference between real axes intercepts of the impedance plot represents the polarization resistance $R_p = R_{HF} + R_{LF}$ [109]. This equivalent circuit yields two circular arcs shifted along the ReZ axis by the amount of $R_{ohm}$. Consequently, the high-frequency intercept on the real axis represents the ohmic resistance of the cell ($R_{ohm}$). The low-frequency intercept on the real axis represents the total cell resistance ($R_{total}$). The difference between the two is the electrode polarization resistance ($R_p$), including the contribution from both the anode and cathode (Figure 3).

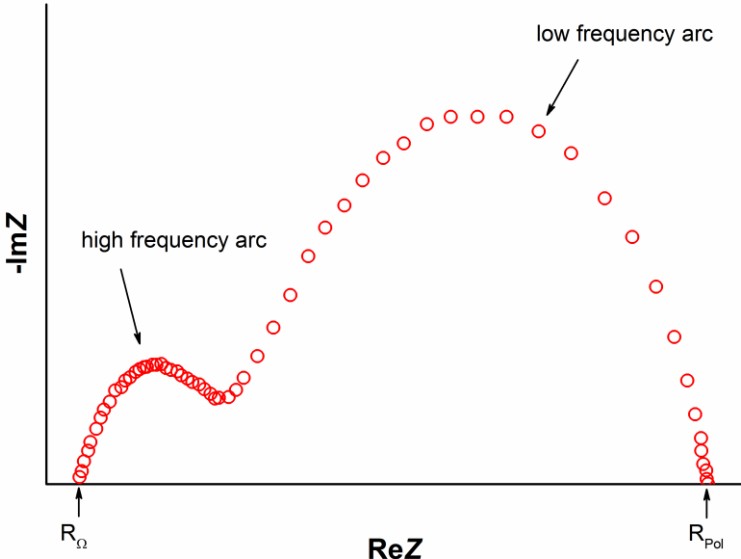

**Figure 3.** EIS spectrum representing the resistances of a SOFC.

Table 7 lists the polarization resistances of operational SOFCs, i.e., including the contribution from both the anode and cathode under their corresponding operational conditions. Values determined under symmetrical atmospheres (without potential chemical gradient) were not considered since the oxygen chemical potential gradient between the cathode and anode of the SOFC and the current across the cell decreased the polarization resistance [17,110]. The lowest $R_p$ value at 800 °C, i.e., 0.152 $\Omega$ cm$^2$, was obtained for an SFM-impregnated YSZ anode in an SFM-YSZ/YSZ/YSZ-LSFSc$_{0.1}$ SOFC configuration [53]. At lower temperatures, the S$_{1.4}$C$_{0.6}$F$_{1.5}$M/LSGM/S$_{1.4}$C$_{0.6}$F$_{1.5}$M SOFC configuration with 35 μm thick electrolytes shows low polarization resistances. Decreasing the electrolyte thickness will further enhance the fuel cell performance [38].

**Table 7.** Polarization resistances of operational SOCFs consisting of SF$_{1+x}$M ceramics.

| Configuration Anode-Electrolyte-Cathode | $T$, °C | $R_p$, $\Omega$ cm$^2$ | Ref. |
|---|---|---|---|
| | 850 | 0.284 | |
| SFM/LSGM/SDC/SmBC$_2$ | 800 | 0.327 | [60] |
| | 750 | 0.583 | |
| SFM-YSZ/YSZ/YSZ-LSFSc$_{0.1}$O | 750 | 0.52 2.11 [1] | [52] |
| SFM-YSZ/YSZ/YSZ-LSFSc$_{0.1}$ SF$_{1.2}$M-YSZ/YSZ/YSZ-LSFSc$_{0.1}$ SF$_{1.35}$M-YSZ/YSZ/YSZ-LSFSc$_{0.1}$ SF$_{1.5}$M-YSZ/YSZ/YSZ-LSFSc$_{0.1}$ | 800 | 0.152 0.176 0.193 0.235 | [52] |

**Table 7.** *Cont.*

| Configuration Anode-Electrolyte-Cathode | $T$, °C | $R_p$, $\Omega$ cm$^2$ | Ref. |
|---|---|---|---|
| SF$_{1.33}$M/LGSM/LSCF | 800 | ~0.2 | [34] |
| | 750 | 0.26 | |
| | 700 | 0.45 | |
| SF$_{1.5}$M/LGSM/LSCF | 850 | 0.40 | [98] |
| | 800 | 0.76 | |
| | 750 | 1.73 | |
| SF$_{1.3}$Co$_{0.3}$M/LSGM/LSCF | 850 | 0.152 | [74] |
| | 800 | 0.223 | |
| | 750 | 0.369 | |
| | 700 | 0.575 | |
| SF$_{1.3}$Co$_{0.3}$M/LSGM/LSCF | 850 | 0.442 [2] | [74] |
| | 800 | 0.842 [2] | |
| | 750 | 1.979 [2] | |
| Ni-LDC/LDC/LGSM/SF$_{1.33}$M | 800 | 0.32 | [101] |
| Ni-YSZ/YSZ/SDC/SF$_{1.5}$M | 750 | 0.42 | [37] |
| | 700 | 0.92 | |
| Ni-YSZ/YSZ/SDC/SF$_{1.5}$M$_{0.4}$N$_{0.1}$ | 800 | 0.168 | [45] |
| SDC-SF$_{1.65}$M/LGSM/SF$_{1.5}$M | 750 | 0.11 0.27 | [111] |
| S$_{1.4}$C$_{0.6}$F$_{1.5}$M/LSGM/S$_{1.4}$C$_{0.6}$F$_{1.5}$M | 800 | 0.155 | [38] |
| | 750 | 0.179 | |
| | 700 | 0.220 | |
| | 650 | 0.359 | |
| SF$_{1.2}$Co$_{0.3}$M/LSGM/SF$_{1.2}$Co$_{0.3}$M | 850 | 0.152 | [74] |
| | 800 | 0.223 | |
| | 750 | 0.369 | |
| | 700 | 0.575 | |
| SF$_{1.5}$M/LSGM/SF$_{1.5}$M$_{0.2}$Sn$_{0.3}$ | 800 | 0.22 | [43] |
| | 750 | 0.35 | |
| | 700 | 0.71 | |

[1] $C_3H_8$, [2] $CH_4$.

Feeding the SOFC with hydrocarbons increases $R_p$ at 800 °C by about a factor of four [52,74]. A similar value at the same temperature was also found in a (PrBa)$_{0.95}$(Fe$_{0.9}$Mo$_{0.1}$)$_2$O$_{5+\delta}$/LSGM/PrBaCo$_2$O$_{5+\delta}$ (PBCO) SOFC configuration when the fuel was switched from H$_2$ to 3% H$_2$O/CH$_4$ [112]. This factor increases with decreasing temperature as a consequence of different activation energies of the involved electrochemical reactions [74].

Generally, the combined area-specific resistivity (ASR) of a SOFC (electrolyte, anode and cathode) should be below 0.5 $\Omega$cm$^2$, ideally approaching 0.1 $\Omega$cm$^2$ to ensure high power densities. Thereby, about 60% is attributed to the electrolyte [1]. Thus, the $R_p$ values listed in Table 7 are still far from the targets.

*4.2. $Sr_2Fe_{1+x}Mo_{1-x}O_{6-\delta}$-Based SOFCs Operated with Hydrogen Fuel*

Initially, $SF_{1.33}M$ and $SF_{1.5}M$ were studied as anodes in parallel [29,34]. A single-phase double perovskite of $SF_{1.33}M$ is formed by treatment in a reducing environment at 800 °C. $SF_{1.33}M$ is compatible with LSGM and a ceria interlayer and shows remarkable electrochemical activity, carbon resistance with $CH_4$ as a fuel and sulfur tolerance in 100 ppm $H_2S/H_2$ [34]. Contrarily to $S_2F_{1.33}M$, $S_2F_{1.5}M$ has already formed a single perovskite phase after the calcination in air. $S_2F_{1.33}M$ shows a better single-cell performance than $SF_{1.5}M$, with $H_2$ and $CH_4$ as fuel attributed to a higher catalytic activity, but it suffers from redox stability [29].

Table 5 compiles the performance of $SF_{1+x}M$-based SOFCs operated with hydrogen fuel. The maximum power of SOFCs using $A_2FeMoO_{6-\delta}$ anodes and $H_2$ as fuel increases in the order: CaFM < BFM < SFM. These SOFCs use a thin CSO buffer layer between the electrolyte and anode to prevent the interdiffusion of ionic species. The poor cell performance of the CaFM anode is attributed to its decomposition at high temperatures and the low oxygen vacancy concentration. SFM exhibited a particularly favorable combination of high electrical conductivity, good thermal stability and thermal compatibility, and electrochemical performance [60]. Neither a composition of $SF_{1+x}M_{1-x}$ [29,34,98] nor B-site substitution [96] significantly changes the SOFC performance. Improvement is achieved by dispersing a small amount of Ni (~2 wt%) on the $SF_{1.5}M0.5$ ceramic anode [99] and for a Sr-deficient anode material [35]. The best performance shows an $SF_{1.4}M$ composition [22], corresponding approximately to the solubility of SMO in $SrFeO_{3-\delta}$ in air at 1200 °C [49,50]. Exsoluted Co nanoparticles boost the $P_{max}$ value [74].

Concerning application as SOFC cathodes, an increase in performance was achieved by A-site substitution of Sr by Ba [37,44], B-site substitution of Fe by Ni [40], Fe by Sc [42], Fe by Co [74], Mo by Nb [45] and by a Sr deficiency [74]. In the case of nearly symmetrical (one of the electrodes modified) SOFCs, a substantial increase in $P_{max}$ can be achieved by the A-site substitution of Sr by Ba at the cathode [38]. A noticeable improvement of $P_{max}$ was demonstrated by a direct comparison of $SF_{1.5}M$ and $SF_{1.4}N_{0.1}M$ anodes [30] and by the application of $SF_{1.5}M_{0.2}Sn_{0.3}$ electrodes [43].

*4.3. $Sr_2Fe_{1+x}Mo_{1-x}O_{6-\delta}$-Based SOFCs Operated with Hydrocarbon Fuel*

The intended fuel of a SOFC is hydrogen. In this case, other fuels, such as natural gas and biomethane, should be first converted to hydrogen in the fuel cell's reformer. With liquid hydrocarbon fueling, SOFCs have essentially the same specific energy as that of the fuel (~1 kWh kg$^{-1}$) [113]. Currently, most hydrogen is produced from fossil fuels, specifically natural gas. The production of $H_2$ requires additional external processes, e.g., steam-methane reforming or partial catalytic oxidation, water splitting by electrolysis, partial combustion of fuel–air or fuel–oxygen mixtures resulting in hydrogen- and carbon monoxide-rich syngas, as well as membrane separation or preferential oxidation. However, each of these steps requires energy. This decreases the overall system efficiency. On the one hand, SOFCs can oxidize essentially any fuel, from hydrogen to hydrocarbons to even carbon, because the electrolyte transports an oxygen ion, $O^{2-}$. On the other hand, any available hydrocarbon-based resource (which includes not only fossil fuels but also, potentially, biomass and municipal solid waste) may be considered as a fuel. Double perovskites have attracted attention due to their beneficial catalytic activity for methane oxidation, reaching 80% at 530 °C [10]. An $SFM_{0.8}N_{0.2}$ anode shows outstanding performances with high resistance against carbon build-up and redox cycling in hydrocarbon fuels [97]. The mixed-valence $Mo^{5+}/Mo^{6+}$ and $Fe^{3+}/Fe^{4+}$ couples provide electronic conductivity, and the characteristic high oxygen vacancy concentration, as well as low oxygen ion migration barriers in SFMO, lead to a high ionic conductivity [33]. The ionic conductivity of SFMO is significantly higher than that of the commonly used $La_{0.8}Sr_{0.2}MnO_3$ cathode, and its electrical conductivity in both air and hydrogen environments meets the conductivity requirements for both anodes and cathodes [29]. The anode materials that operate directly with hydrocarbon fuels should be chemically stable in the $CO_2$ atmosphere because

this is the main oxidation product formed in the anode compartment. $SrF_{1.5}M$ is stable both under $H_2$ and pure $CO_2$ atmospheres [17]. SOFCs are able to convert the energy stored in any hydrocarbon fuel directly into electricity. However, sulfur species in the available hydrocarbon fuels will unfavorably react with the Ni catalyst in conventional nickel-yttrium-stabilized zirconia (Ni-YSZ) anode, resulting in severe anode sulfur poisoning and, thus, significant cell performance loss. $SF_{1+x}M$ electrodes show an excellent low level of sulfur poisoning [31,95,114]. From all these points of view, half-metallic double-perovskites with a double-exchange conduction mechanism seem to be perfect candidates in fuel cells using hydrocarbons or alcohols as fuel. Table 6 compiles the performance of $SF_{1+x}$M-based SOFCs operated with hydrogen fuel.

The performance of $SF_{1+x}$M-based SOFCs with $CH_4$ and methanol as fuels is comparable with the state-of-the-art [115]. For instance, in [116], a novel on-cell micro-reformer, i.e., a noble-metal free $NiSn/Al_2O_3$ catalyst embedded Ni foam, was designed for efficient internal reforming of biogas in SOFCs. The NiSn bimetallic alloys were developed and characterized as the potential anode material and biogas reforming catalyst. Sn remained preferentially on the surface of the NiSn alloy to enhance both the carbon deposition resistance and sulfur tolerance $NiSn/Al_2O_3$ deposited on Ni foam and functioned as a catalyst layer, which proved with high catalytic activity as well as great stability toward biogas reforming reactions. In the button cell with the configuration $NiSn/Al_2O_3 \,|\, NiSn–Y_{0.08}Zr_{0.92}O_2 \,|\, Y_{0.08}Zr_{0.92}O_2 \,|\, Y_{0.08}Zr_{0.92}O_2–(La_{0.8}Sr_{0.2})_{0.95}MnO_3$, the peak power density of $0.946 \text{ W/cm}^2$ was achieved at 850 °C with the inlet gas of $CH_4–CO_2–200$ ppm $H_2S$, and conversion rate to methane reached around 95% with the discharge current density of $1.25 \text{ A/cm}^2$. This performance is comparable with the data in Table 6.

## 5. Conclusions

In this work, we reviewed the application of $Sr_2FeMoO_{6-\delta}$ (SFM) and $Sr_2Fe_{1-x}Mo_{1-x}O_{6-\delta}$ ($SF_{1+x}M$) in LSGM-based SOFCs. These materials are stable in both oxidizing and reducing atmospheres enabling the use of both not only as an anode but also as a cathode and also in symmetrical SOFCs. They provide both electron and ionic conductivity extending the triple phase boundary compared to purely electronic or purely ionic conductors. Their properties can be tailored to a particular application by the substitution of different metal cations into their lattices. $SF_{1+x}M$ materials are excellent catalysts in hydrocarbon oxidation and can prevent carbon deposition due to the ability to exchange lattice oxygen with the gaseous phase. Moreover, they are sulfur tolerant. This opens the way to direct hydrocarbon-fueled SOFCs eliminating the need for external fuel reforming and sulfur removal components. As such, SOFCs can be greatly simplified and operate with much higher overall efficiency, thus contributing to the solution to the lack of energy problem in our modern world.

Nevertheless, there are still some problems that need to be solved for commercial application: (i) A vapor phase deposition technology for SF1+xM deposition with a reproducible composition has to be developed. (ii) The composition of an SF1+xM cathode should be adapted to the used electrolyte. It will be different in the case of LSGM and YSZ electrolytes. (iii) A vapor phase deposition method of porous, composite anode deposition is required. Catalytic properties may be further improved by impregnation and exsolution methods. (iv) An appropriate carrier material of the SOFC thermally matched to the electrodes and the electrolyte must be selected.

**Author Contributions:** Conceptualization, G.S.; methodology, G.S.; software, E.A.; validation, G.S., and E.A.; formal analysis, E.A.; investigation, E.A.; resources, G.S.; data curation, E.A.; writing—original draft preparation, G.S.; writing—review and editing, E.A.; visualization, E.A.; supervision, G.S.; project administration, G.S.; funding acquisition, G.S. All authors have read and agreed to the published version of the manuscript.

**Funding:** This research was funded by the European Union within the scope of the European project H2020-MSCA-RISE-2017-778308–SPINMULTIFILM.

**Institutional Review Board Statement:** Not applicable.

**Informed Consent Statement:** Not applicable.

**Data Availability Statement:** The original contributions presented in the study are included in the article; further inquiries can be directed to the corresponding author.

**Acknowledgments:** The authors have benefited from comments and valuable discussions with N. Sobolev (University Aveiro).

**Conflicts of Interest:** The authors declare no conflict of interest.

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
