# Peer review of "Nonstoichiometric Strontium Ferromolybdate as an Electrode Material for Solid Oxide Fuel Cells"

_inorganics, doi:10.3390/inorganics10120230_

Round 1
Reviewer 1 Report
Use of abbreviations for makeing the article more reader friendly: When the author choose to use short forms of the compositions (e.g. SF1+xM), please check that the full formula is written the first time the abbreviation is used. (check line 136 and 141). Please ensure one letter is only used for one elment, thus C = Co, M = Mo, S = Sr. Thus use Ce, Cr, Ca, Sn, Sm, Sc when these elements are occuring. The exceptions are for the very common materials LSGM and CGO.
Line 269: SFMO is not logical
Line 318: Full name of DFT required
Section reagarding electrical conductivity: I suggest you only refere to electrical conductivity which is either increasing or decreasing with increasing temperature. When the also electrical resistivity is used, the understanding becomes more difficult. (line 448 and other places)
Can you also include the absolute conductivity in table 4.
Table 5 and 6: Please add some horizontal lines between each group of rows where every coloumn have new values.
Table 5: What is the Pr?
Line 777: This sound strange, I suggest "The typical fuel of a SOFC in electrode development experiment is hydrogen" Or perhaps some other reformulation
Line 811: which state-off-the-ar
Minor suggestion / typos
Line 395: That is repeated
Line 42: Paranthesis around (O2).
Line 551: "and" instead of "but"?
Line 648: "not any"
Author Response
Use of abbreviations for making the article more reader friendly: When the author choose to use short forms of the compositions (e.g. SF1+xM), please check that the full formula is written the first time the abbreviation is used. (check line 136 and 141). Please ensure one letter is only used for one element, thus C = Co, M = Mo, S = Sr. Thus use Ce, Cr, Ca, Sn, Sm, Sc when these elements are occuring. The exceptions are for the very common materials LSGM and CGO.
We have now generalized the notations. All designations were introduced by chemical formulas. Moreover, we have added a sentence summarizing all abbreviations.
After revision, only one letter is used for one element:
B – Bax, Ba1-x, Ba2-x introduced by Ba0.5Sr0.5Co0.8Fe0.2O3-δ (BSCF), Ba2FeMoO6−d (B2FMo), Ba0.5Sr0.5Co0.8Fe0.2O3-δ (BSCF), Ba0.9Co0.7Fe0.2Nb0.1O3-δ (BCFN0.1),
C – Cax, introduced by Ca2FeMoO6−d (CFM), Sr2-xCaxFe1.5MoO6-d (S2-xCxFM),
Co – Co1-x, Co2-x, introduced by La0.8Sr0.2Fe0.8Co0.2O3-δ (LSCF), La0.8Sr0.2Fe0.8Co0.2O3-δ (LSCF), SmBaCo2O5+x (SmBC2O),
Ce - Ce1-x, introduced by Ce1-xLaxO2-d (LDC) Ce1-xGdxO2-d, (GDC), Ce1-xSmxO2- d, (SDC),
F – Fe1+x, introduced by Sr2FeMoO6-d (SFM), SrFe1.5Mo0.5O6-d (SF1.5Mo),
G-Ga1-x, M- Mgx introduced by La1-xSrxGa1-yMgyO3-δ (LSGM),
Gd – Gdx introduced by Ce1-xGdxO2-d (GDC),
L - Lax, La1-x, introduced by Ce1-xLaxO2-d (LDC), La0.8Sr0.2Fe0.8Co0.2O3-δ (LSCF), La1-xSrxMnyO3-δ (LSMn), and La0.6Sr0.4Fe0.9Sc0.1O3-δ (LSFSc0.1),
M – Mo introduced by Sr2FeMoO6-d (SFM), Sr2F1.+xMo1-xO6-d (SF1+xM)
Mn – Mnx, introduced by La0.8Sr0.2MnO3 (LSMn),
Mo - Mox, Mo1-x, introduced by Sr2FeMoO6-d (SFM), SrFe1.5Mo0.5O6-d (SF1.5M),
N – Nbx, introduced by Ba0.9Co0.7Fe0.2Nb0.1O3-δ introduced by (BCFN), Sr2Fe1.5Mo0.5-xNbxO6-d (SF1.5M0.5-xNx),
Ni – Nix, Sr2Fe1.4Ni0.1Mo0.5O6-d (SF1.4Ni0.1M) and Sr1.9Fe1.4Ni0.1Mo0.5O6-d (S1.9F1.4Ni0.1M)
O – On-d (n= 2,3,6) introduced by Ce1-xLaxO2-d (LDC), Ce1-xGdxO2-d (GDC), La1-xSrxGa1-yMgyO3-δ (LSGM), La0.8Sr0.2Fe0.8Co0.2O3-δ (LSCF), Ba0.5Sr0.5Co0.8Fe0.2O3-δ (BSCF) Sr2FeMoO6-d (SFM), SrFe1.5Mo0.5O6-d (SF1.5M), Ba2FeMoO6−d, Ca2FeMoO6−d (CFM)
P – Pr, introduced by PrBaCo2O5+δ(PBCO),
S - Srx, Sr1-x, Sr2-x, introduced by La1-xSrxMnyO3-δ (LSMn), Sr2FeMoO6-d (SFM), SrFe1.5Mo0.5O6-d (S2F1.5M),
Sc – Scx, introduced by La0.6Sr0.4Fe0.9Sc0.1O3-δ (LSFSc0.1)
Sm – Smx introduced by SmBaCo2O5+x (SmBC2O), Ce1-xSmxO2- d, (SDC)
O – On-d (n= 2,3,6) introduced by Sr2FeMoO6-d (SFM), SrFe1.5Mo0.5O6-d (SF1.5M),
Sc - Scx, introduced by La0.6Sr0.4Fe0.9Sc0.1O3-δ (LSFSc0.1)
Sm -Sm introduced by SmBaCo2O5+x (SmBC2),
Y – Y, introduced by Y1-xZrxO2-d (YSZ)
Z – Zr, introduced by Y1-xZrxO2-d (YSZ)
We have added a sentence in row 50 to clear the notation for the reader.
Line 269: SFMO is not logical
Is now noted as SFM
Line 318: Full name of DFT required
Done and explained.
Section regarding electrical conductivity: I suggest you only refer to electrical conductivity which is either increasing or decreasing with increasing temperature. When the also electrical resistivity is used, the understanding becomes more difficult. (line 448 and other places)
We refer to the transport of charge carriers subjected at different temperatures to different mechanisms, and we show that in the area of interest is such a change from adiabatic polaron hopping to a reduction of charge carriers by oxygen vacancy formation.
Can you also include the absolute conductivity in table 4.
The physical parameters in this case ore the pre-exponential factors at low temperatures which might be given, but absolute values of conductivity were already given in the text. Note that ionic conductivity is negligible compared to the electronic one.
Table 5 and 6: Please add some horizontal lines between each group of rows where every coloumn have new values.
done
Table 5: What is the Pr?
This was a remaining misprint.
Line 777: This sound strange, I suggest "The typical fuel of a SOFC in electrode development experiment is hydrogen" Or perhaps some other reformulation.
We have changed the text “The intended fuel of a SOFC” and have added: “In this case, other fuels like natural gas and biomethane should be first converted to hydrogen in the fuel cell’s reformer” for explanation.
Line 811: which state-off-the-ar
This should be a remaining misprint. In my file “state-of-the art” is written
Minor suggestion / typos
Line 395: That is repeated
corrected
Line 42: Paranthesis around (O2).
done
Line 551: "and" instead of "but"?
Corrected into “and not along” which seems to be more appropriate.
Line 648: "not any"
Corrected into “no degradation”
Reviewer 2 Report
Dear authers,
On one hand the review is a great compilaion of studies in the field and as such can be useful as a reference. On the other hand, it does not tell a story or a narative about the subject leading it to be a poor read other than the collected tables.
some specific comments, notice that I have stopped checking grammer and english soon since it was takeing too much of my time.
Review for:
Nonstoichiometric strontium ferromolybdate as an electrode material for solid oxide fuel cells
This work is written in poor grammar, many sentences are incomplete, there are jumps between subjects and inconsistencies. Please send to an English Editor before review. I have started correcting some mistakes but if I kept going the review would be 30 pages long.
“These materials are stable in both oxidizing and reducing atmospheres enabling the use of both not only as an anode, but also as a cathode and also in symmetrical SOFCs.”
This sentence appears twice in the abstract:
Line 60 in page 2
Sr2+ doping is carried out at the La3+ site to introduce in the parent structure of LaMnO3 oxygen vacancies due to the charge compensation mechanism
This is not true, Sr creates Mn4+ which creates mobile holes, no oxygen vacancies are created which is the reason LSM does not conduct oxygen.
It is unclear what is the problem with LSCF.
You continuously jump between electrolytes (YSZ, Ceria, LSGM) without being clear why are you talking about each. One cathode is for YSZ, the next is for CGO the next LSGM without a clear jump in-between.
Towards lines 70-75 in page 2 it’s just a list of cathodes without a narrative. Why are they not what you are writing about them? You sort of give a list of cathodes without ending with a clear and concise reason why SFM is the subject of the review.
In 77-92 in page 2 the electrolytes are referenced after optional cathodes are explained however the cathodes are for specific electrolytes (as I have explained before). I would switch between the cathodes and the electrolytes.
Line 93- The considered [cathode? Anode? Material?] in this work.
93-99 you explain for LSGM the testing conditions (no shift in peak positions) where as with CGO and YSZ it’s just “is stable” there is no reason to do the differentiation. It’s stable with all 3. Also with LSGM the lower temperature is mentioned (800) but not for YSZ or CGO.
115 – the first explanation of a symmetrical SOFC is presented after a material is being discussed. I think discussing this as a goal in the beginning of the review would make the case for SFM much stronger. It is also worth noting the symmetrical nature of thermal expansion in such cells.
Figure 2, why use a schematic figure and not actual numbers in a specific example?
Line 825 – in this work the application of LFM in LSGM SOFC’s – this is not a complete sentence.
The conclusions don’t separate between what is solved and known and what still needs to be improved. Basically, the conclusions are LFM is good. OK but what is it better at and what does it still need to be improved in? Is it to the point where it should replace commercial cathodes and anodes?
Author Response
This work is written in poor grammar, many sentences are incomplete, there are jumps between subjects and inconsistencies. Please send to an English Editor before review. I have started correcting some mistakes but if I kept going the review would be 30 pages long.
The work is now corrected by a native speaker.
“These materials are stable in both oxidizing and reducing atmospheres enabling the use of both not only as an anode, but also as a cathode and also in symmetrical SOFCs.”
This sentence appears twice in the abstract:
Now deleted.
Line 60 in page 2
Sr2+ doping is carried out at the La3+ site to introduce in the parent structure of LaMnO3 oxygen vacancies due to the charge compensation mechanism. This is not true, Sr creates Mn4+ which creates mobile holes, no oxygen vacancies are created which is the reason LSM does not conduct oxygen.
In the case of aliovalent doping in the A-site, the electrical neutrality of the system can be compensated either by changing the oxidation state of multivalent cations at B-sites or by formation of lattice oxygen vacancies. Both mechanisms are discussed for LaMnO3 in literature. Sr2+-doping has been used for La3+ substitution to enhance electrical conductivity. Indeed, the strontium (valence 2+) doping on lanthanum (valence 3+) introduces extra holes in the valence band and thus increases electronic conductivity. On the other hand, the review [https://link.springer.com/article/10.1007/s10008-009-0932-0] discusses the promotion of oxygen vacancy formation replacing La3+ by Sr2+. Here, the increase of the ionic conductivity with Sr-doping give evidence of oxygen vacancy formation. We thank the referee for this hint and have now added the fact of the increase of charge carriers in the p-type material to the manuscript text for completeness.
It is unclear what is the problem with LSCF.
A sufficient problem off all cobalt containing SOFC electrode materials is the relatively high value of the thermal expansion coefficient [https://doi.org/10.1016/j.ijhydene.2016.04.097]. This can be avoided by A-site deficiency. However, this is hard to realize by vapor phase deposition methods. Here sol-gel technology could be a solution, but not for large areas.
You continuously jump between electrolytes (YSZ, Ceria, LSGM) without being clear why are you talking about each. One cathode is for YSZ, the next is for CGO the next LSGM without a clear jump in-between.
The focus of this review is strontium ferromolybdate. Correspondingly we consider how it is adapted to various electrolytes. Only YSZ and LSGM were used as electrolytes for SOFCs comprising SrFe1+xMo1-xO6-d electrodes. YSZ was selected due to its chemical stability, LSGM due to a higher ionic conductivity. The ceria interlayer prevent reactions between a Ni-containing anode and the LSGM electrolyte. Correspondingly, only these items were included in our review paper.
Towards lines 70-75 in page 2 it’s just a list of cathodes without a narrative. Why are they not what you are writing about them? You sort of give a list of cathodes without ending with a clear and concise reason why SFM is the subject of the review.
That part of the manuscript text introduces solely that cathode materials which were applied previously in SrFe1+xMo1-xO-based SOFCs (cf. table 6 and 7) in order to explain the abbreviations in tables 6 and 7.
In 77-92 in page 2 the electrolytes are referenced after optional cathodes are explained however the cathodes are for specific electrolytes (as I have explained before). I would switch between the cathodes and the electrolytes.
The aim of this work was to show that Sr2Fe1+xMo1-xO6-d is a suitable cathode material in connection with LSGM and YSZ electrolytes and not to adapt a cathode material to a given electrolyte.
Line 93- The considered [cathode? Anode? Material?] in this work.
We do not give recommendations on the optimized SOFC. Instead, we discuss the advantages and drawbacks of all materials reported up to now in Sr2Fe1+xMo1-xO6-d-based SOFCs. All the materials have specific drawbacks discussed to a certain extent in this work. An application-related tradeoff should be found by the reader.
93-99 you explain for LSGM the testing conditions (no shift in peak positions) whereas with CGO and YSZ it’s just “is stable” there is no reason to do the differentiation. It’s stable with all 3. Also with LSGM the lower temperature is mentioned (800) but not for YSZ or CGO.
To the best of our knowledge, we have not find a literature report proving the chemical stability of LSGM with SrFe1+xMo1-xO electrodes. Only one report communicates no shift in XRD peak position that we have adopt correspondingly. However, in our opinion, this is not a convincing proof. The facts are described as they are. The assessment we address now to the responsibility of the skilled reader.
115 – the first explanation of a symmetrical SOFC is presented after a material is being discussed. I think discussing this as a goal in the beginning of the review would make the case for SFM much stronger. It is also worth noting the symmetrical nature of thermal expansion in such cells.
This break is now shifted now before material consideration. The advantage of balanced stress was added.
Figure 2, why use a schematic figure and not actual numbers in a specific example?
Perovskites suitable for SOFC electrode application show a similar conductivity behavior with a change of the dominant conductivity mechanism from adiabatic small polaron hopping to carrier diminution caused by oxygen lattice loss. Such a common behavior is best illustrated by a schematic figure based on real experimental data.
Line 825 – in this work the application of LFM in LSGM SOFC’s – this is not a complete sentence.
Some text was missing. It should be “In this work, we reviewed the application of Sr2FeMoO6-d (SFM) and Sr2Fe1-xMo1-xO6-d (SF1+xM) in LSGM-based SOFCs.”.
The conclusions don’t separate between what is solved and known and what still needs to be improved. Basically, the conclusions are LFM is good. OK but what is it better at and what does it still need to be improved in? Is it to the point where it should replace commercial cathodes and anodes?
The promising features of SF1+xM were already given in the conclusions. Now, we have added also some problems to be solved before commercialization.
Round 2
Reviewer 2 Report
While my points we adressed the main point of the paper not being clear, concice or telling a coherent story had not been. The paper still feels disjointed.